# Sorting with Predictions

**Xingjian Bai**
Department of Computer Science
University of Oxford, UK
xingjian.bai@sjc.ox.ac.uk

**Christian Coester**
Department of Computer Science
University of Oxford, UK
christian.coester@cs.ox.ac.uk

## Abstract

We explore the fundamental problem of sorting through the lens of learning-augmented algorithms, where algorithms can leverage possibly erroneous predictions to improve their efficiency. We consider two different settings: In the first setting, each item is provided a prediction of its position in the sorted list. In the second setting, we assume there is a "quick-and-dirty" way of comparing items, in addition to slow-and-exact comparisons. For both settings, we design new and simple algorithms using only $O(\sum_i \log \eta_i)$ exact comparisons, where $\eta_i$ is a suitably defined prediction error for the $i$th element. In particular, as the quality of predictions deteriorates, the number of comparisons degrades smoothly from $O(n)$ to $O(n \log n)$. We prove that this comparison complexity is theoretically optimal with respect to the examined error measures. An experimental evaluation against existing adaptive and non-adaptive sorting algorithms demonstrates the potential of applying learning-augmented algorithms in sorting tasks.

## 1 Introduction

Sorting is one of the most basic algorithmic problems, commonly featured as one of the initial topics in computer science education, and with a vast array of applications spanning various domains. In recent years, the emerging field of algorithms with predictions (Lykouris and Vassilvitskii, 2021, Mitzenmacher and Vassilvitskii, 2022), also known as learning-augmented algorithms, has opened up new possibilities for algorithmic improvement, where algorithms aim to leverage predictions (possibly generated through machine learning, or otherwise) to improve their performance. However, the classical sorting problem with predictions, along with the discussion of different types of predictors for sorting, appears to have been largely overlooked by this recent movement. This paper explores the problem of sorting through the lens of algorithms with predictions in two settings, aiming to overcome the classical $\Omega(n \log n)$ barrier with the aid of various types of predictors.

The first setting involves each item having a prediction of its position in the sorted list. This type of predictor is commonly found in real-world scenarios. For instance, empirical estimations of element distribution can generate positional predictions. Another example is that a fixed set of items has their ranking evolve over time, with minor changes at each timestep. Here, an outdated ranking can serve as a natural prediction for the current ranking. As re-evaluating the true relation between items can be costly, the provided positional predictions offer useful information. The positional prediction setting is closely related to *adaptive sorting* of inputs with existing presortedness (Estivill-Castro and Wood, 1992), but we consider different measures of error on the predictor, resulting in algorithms with a more fine-grained complexity.

In the second setting, a "dirty" comparison function is provided to assist sorting. In biological experiments, for instance, some "indicating factors" might be used to approximately compare two molecules or drugs. Despite potential errors due to the oversight of minor factors, these comparisons can still offer preliminary insights into the properties of the subjects. More broadly, in experimental science, researchers often carry out costly experiments to compare subject behaviours. By utilizing a

37th Conference on Neural Information Processing Systems (NeurIPS 2023).

proficient sorting algorithm that capitalizes on dirty comparisons, the need for costly experiments can be reduced and substituted by less expensive, albeit noisier, experiments.

We propose sorting algorithms to leverage either type of predictor. In the positional prediction setting, we design two deterministic algorithms with different complexity bounds, while in the dirty comparisons setting, we develop a randomized algorithm. In all settings, we provide bounds of the form $O(\sum_{i=1}^{n} \log(\eta_i + 2))$ on the number of exact comparisons, for different notions of element-wise prediction errors $\eta_i \in [0, n]$. In particular, all three proposed algorithms only require $O(n)$ exact comparisons if predictions are accurate (*consistency*), never use more than $O(n \log n)$ comparison regardless of prediction quality (*robustness*), and their performance degrades slowly as a function of prediction error (*smoothness*). Moreover, we show that all algorithms have optimal comparison complexity with respect to the error measures examined.

Finally, through experiments on both synthetic and real-world data, we evaluate the proposed algorithms against existing (adaptive and non-adaptive) algorithms. Results demonstrate their superiority over the baselines in multiple scenarios.

## 1.1 Preliminaries

Let $A = \langle a_1, \ldots, a_n \rangle$ be an array of $n$ items, equipped with a strict linear order $<$. Let $p \colon [n] \to [n]$ be the permutation that maps each index $i$ to the position of $a_i$ in the sorted list; that is, $a_{p^{-1}(1)} < a_{p^{-1}(2)} < \cdots < a_{p^{-1}(n)}$. We consider two settings of sorting with predictions.

**Sorting with Positional Predictions.** In *sorting with positional predictions*, the algorithm receives for each item $a_i$ a prediction $\hat{p}(i)$ of its position $p(i)$ in the sorted list. We allow $\hat{p}$ to be any function $[n] \to [n]$, which need not be a permutation (i.e., it is possible that $\hat{p}(i) = \hat{p}(j)$ for some $i \neq j$).

Positional predictions can be generated by models that roughly sort the items, e.g. focusing on major factors while neglecting minor ones. Or they can stem from past item rankings, while the properties of the items evolve over time. In such cases, the objective is to obtain the latest ranking of items.

The error of a positional prediction can be naturally quantified by the displacement of each element's prediction; that is, the absolute difference of the predicted ranking and the true ranking. We define the *displacement error* of item $a_i$ as

$$\eta_i^{\Delta} := |\hat{p}(i) - p(i)|.$$

The following notion of one-sided error provides an alternative perspective to evaluate the complexity of algorithms with positional predictions. We denote the left-error and right-error of item $a_i$ as

$$\eta_i^l := |\{j \in [n] \colon \hat{p}(j) \leq \hat{p}(i) \wedge p(j) > p(i)\}|$$
$$\eta_i^r := |\{j \in [n] \colon \hat{p}(j) \geq \hat{p}(i) \wedge p(j) < p(i)\}|.$$

In certain contexts, it may be impossible to obtain a predictor with small displacement error, but possible to obtain one with a small one-sided error. By developing algorithms for this setting, we expand the space of problems where sorting algorithms with predictions can be applied.

**Sorting with Dirty and Clean Comparisons.** The other setting we consider involves a predictor that estimates which of two elements is larger without conducting a proper comparison, providing a faster but possibly inaccurate result. This type of predictor is applicable in scenarios where exact comparisons are costly but a rough estimate of the comparison outcome can be obtained more easily.

Formally, in *sorting with dirty comparisons*, the algorithm has access to a complete, asymmetric relation $\widehat{<}$ on the items, while still also having access to the exact comparisons $<$. That is, for any two distinct items $a_i$ and $a_j$, either $a_i \widehat{<} a_j$ or $a_j \widehat{<} a_i$. We think of $\widehat{<}$ as an unreliable, but much faster to evaluate prediction of $<$. We also refer to the (fast) comparisons according to $\widehat{<}$ as *dirty* whereas the (slow) comparisons according to $<$ are *clean*. We emphasize that the relation $\widehat{<}$ need *not be transitive*, so it is not necessarily a linear order. Instead, $\widehat{<}$ induces a tournament graph on the items of $A$, containing a directed edge $(a_i, a_j)$ if and only if $a_i \widehat{<} a_j$, and in many applications, we expect this graph to have cycles.

We denote by $\eta_i$ the number of incorrect dirty comparisons involving $a_i$, that is,

$$\eta_i := \left| \{j \in [n] \colon (a_i < a_j) \neq (a_i \widehat{<} a_j)\} \right|.$$

## 1.2 Main Results

Our main result for the dirty comparison setting is given by the following theorem:

**Theorem 1.1.** *Augmented with dirty comparisons, there is a randomized algorithm that sorts an array within $O(n \log n)$ running time, $O(n \log n)$ queries to dirty comparisons, and $O\left(\sum_{i=1}^{n} \log\left(\eta_i + 2\right)\right)$ clean comparisons in expectation.*

By the classical lower bound, the total number of dirty + clean comparisons must be at least $\Omega(n \log n)$ regardless of prediction quality, but for sufficiently good predictions the theorem allows us to replace most clean comparisons with dirty ones. The case where dirty comparisons are probabilistic is discussed in Section B.1.

The following theorem generalizes the previous one to the case that there are $k$ different dirty comparison operators:

**Theorem 1.2.** *Augmented with $k \leq 2^{O(n/\log n)}$ dirty comparison predictors, where the error of the $p$th predictor is denoted by $\eta^p$, there is a randomized algorithm that sorts an array with at most $O(\min_p \sum_{i=1}^{n} \log(\eta_i^p + 2))$ clean comparisons.*

In other words, the number of clean comparisons is as good as if we knew in advance which of the $k$ predictors is best. The bound on $k$ is almost tight, since already $k \geq 2^{n \log n}$ would mean there could be one predictor for each of the $n!$ possible sorting outcomes, which would render them useless. The proof of Theorem 1.2 is given in Appendix B.2.

The next two theorems capture our algorithms for the positional prediction setting:

**Theorem 1.3.** *Augmented with a positional predictor, there is a deterministic algorithm that sorts an array within $O\left(\sum_{i=1}^{n} \log\left(\eta_i^\Delta + 2\right)\right)$ running time and comparisons.*

**Theorem 1.4.** *Augmented with a positional predictor, there is a deterministic algorithm that sorts an array within $O\left(\sum_{i=1}^{n} \log\left(\min\left\{\eta_i^l, \eta_i^r\right\} + 2\right)\right)$ comparisons.*

We remark that there exist instances where the bound of Theorem 1.3 is stronger than that of Theorem 1.4 and vice versa.[1]

The following theorem, proved in Appendix E, shows tightness of the aforementioned upper bounds.

**Theorem 1.5.** *Augmented with dirty comparisons, no sorting algorithm uses $o\left(\sum_{i=1}^{n} \log\left(\eta_i + 2\right)\right)$ clean comparisons. Augmented with a positional predictor, no sorting algorithm uses $o\left(\sum_{i=1}^{n} \log\left(\eta_i^\Delta + 2\right)\right)$ or $o\left(\sum_{i=1}^{n} \log(\min\left\{\eta_i^l, \eta_i^r\right\} + 2)\right)$ comparisons.*

**Bounds in Terms of Global Error.** One may wonder how the above bounds translate to a global error measure such as the number of item pairs where the larger one is incorrectly predicted to be no larger than the smaller one. Writing $D$ for this error measure, in the dirty comparison setting we simply have $D = \frac{1}{2}\sum_i \eta_i$, and in the positional prediction setting $D \geq \frac{1}{2}\sum_i \eta_i^\Delta$ by (Rohatgi, 2020, Lemma 11). Thus, concavity of logarithm and Jensen's inequality yield an upper bound of $O\left(n \log\left(\frac{D}{n} + 2\right)\right)$ for both settings. This bound is tight[2] as a function of $D$ and corresponds to the optimal complexity of adaptive sorting as a function of the number of inversions (Mannila, 1985).

However, our guarantees in terms of element-wise error are strictly stronger whenever Jensen's inequality is not tight, i.e., when the $\eta_i$ are non-uniform. Furthermore, it is reasonable to expect predictors to exhibit varying levels of error for different items, especially when the error originates from element-wise noise.

## 1.3 Related Works

**Algorithms with Predictions.** Our study aligns with the broader field of learning-augmented algorithms, also known as algorithms with predictions. The majority of research has focused on

---

[1]If predictions are correct except that the positions of the $\sqrt{n}$ smallest and $\sqrt{n}$ largest items are swapped, then $\sum_i \log(\eta_i^\Delta + 2) = \Theta(n)$, but $\sum_i \log(\min\{\eta_i^l, \eta_i^r\} + 2)) = \Theta(n \log n)$. Conversely, if $\hat{p}(i) = p(i) + \sqrt{n}$ mod $n$, then $\sum_i \log(\eta_i^\Delta + 2) = \Theta(n \log n)$, but $\sum_i \log(\min\{\eta_i^l, \eta_i^r\} + 2)) = \Theta(n)$.

[2]Indeed, our proof of Theorem 1.5 constructs a family of instances where each $\eta_i$ is bounded by the same quantity, so Jensen's inequality is tight for these instances.

classical online problems such caching (Lykouris and Vassilvitskii, 2021, Rohatgi, 2020, Wei, 2020, Bansal et al., 2022), rent-or-buy problems (Purohit et al., 2018, Gollapudi and Panigrahi, 2019, Angelopoulos et al., 2020, Wang et al., 2020, Antoniadis et al., 2021), scheduling (Purohit et al., 2018, Lattanzi et al., 2020, Mitzenmacher, 2020, Azar et al., 2021, 2022, Lindermayr and Megow, 2022) and many others. In comparison, research on learning-augmented algorithms to improve runnning time for offline problems is relatively sparser, but examples include matching (Dinitz et al., 2021, Sakaue and Oki, 2022), clustering (Ergun et al., 2022), and graph algorithms (Chen et al., 2022, Davies et al., 2023). Motivated by (Kraska et al., 2018), (Lykouris and Vassilvitskii, 2021) describes a simple method to speed up binary search with predictions, which has been inspirational for our work. Learning-augmented algorithms have also been applied to data structures such as binary search trees (Lin et al., 2022, Cao et al., 2023), and empirical works demonstrate the benefits of ML-augmentation for index structures (Kraska et al., 2018) and database systems (Kraska et al., 2019). There has also been increasing interest in settings where algorithms have access to multiple predictors (Gollapudi and Panigrahi, 2019, Wang et al., 2020, Bhaskara et al., 2020, Almanza et al., 2021, Emek et al., 2021, Dinitz et al., 2022, Anand et al., 2022, Antoniadis et al., 2023).

Related to sorting, Lu et al. (2021) studied learning-augmented *generalized* sorting, a variant of sorting where some comparisons are allowed while others are forbidden. The predictions they consider are similar to our dirty comparisons. They proposed two algorithms with comparison complexities $O(n \log n + w)$ and $O(nw)$, where $w$ is the total number of incorrect dirty comparisons. In the classical (non-generalized) setting with all comparisons allowed, only the second bound theoretically improves upon $O(n \log n)$, but the dependence on the error is exponentially worse than for our algorithms; even with a single incorrect dirty comparison per item, the $O(nw)$ bound becomes $O(n^2)$, whereas ours is $O(n)$. A recent work of Erlebach et al. (2023) studies sorting under explorable uncertainty with predictions, where initially only an interval around the value of each item is known, the exact values can be queried, a prediction of these values is given, and the goal is to minimize the number of queries needed to sort the list.

**Deep Learning-Based Sorting.** The thriving development of deep learning has inspired research into new sorting paradigms. A DeepMind study (Mankowitz et al., 2023) recast sorting as a single-player game, training agents to perform effectively, thus uncovering faster sorting routines for short sequences. Kristo et al. (2020) proposed a sorting algorithm that uses a learning component to improve the empirical performance of sorting numerical values. Their algorithm tries to approximate the empirical CDF of the input by applying ML techniques to a small subset of the input. This setting diverges from ours: If inputs are non-numeric (and no monotonous mapping to numbers is known), then one has to rely on a comparison function, and the approach of Kristo et al. (2020) would not be well-defined, whereas our algorithms can sort arbitrary data types.

Conversely, the input in (Kristo et al., 2020) is solely the item list, without any additional predictions. Note that predictions (or other assumptions) are necessary to beat the entropic $\Omega(n \log n)$ lower bound. [3]

**Noisy Sorting.** Noisy sorting contemplates scenarios where comparison results may be incorrect. This model is useful to simulate potential faults in large systems. Two noisy sorting settings have primarily been considered: In *independent* noisy setting, each query's result is independently flipped with probability $p \in (0, \frac{1}{2})$. Recently, Gu and Xu (2023) provided optimal bounds on the number of queries to sort $n$ elements with high probability. *Recurrent* noisy setting (Braverman and Mossel, 2008) further assumes any repeated comparisons will yield consistent results. Geissmann et al. (2019) present an optimal algorithm that guarantees $O(n \log n)$ time, $O(\log n)$ maximum dislocation, and $O(n)$ total dislocation with high probability. While the recurrent noisy setting is closely related to dirty comparisons, studies in that field focus primarily on approximate sorting; to the best of our knowledge, no *exact* sorting algorithms that use both dirty and clean comparisons have been studied.

**Adaptive Sorting.** Adaptive sorting algorithms leverage various types of existing order in the input, achieving better complexity for partially sorted data. Examples include TimSort (Peters, 2002), which is the standard sorting algorithm in a variety of programming languages, Cook-Kim division

---

[3]The theoretical guarantee of their algorithm is $O(n^2)$, although they observe much better empirical performance. Indeed, this makes sense for numerical inputs drawn from a sufficiently nice distribution, since then one can extrapolate from a small part of the input to the rest.

(Cook and Kim, 1980), and Powersort (Munro and Wild, 2018), which recently replaced TimSort in Python's standard library. We refer to the survey of Estivill-Castro and Wood (1992) for a broader overview. The concept of pre-sortedness is closely related to positional predictions; however, without the motivation from predictors, the complexity bound on adaptive sorting algorithms was often considered under error measures on the entire array, instead of on each element. In contrast, our error measure is element-wise, allowing algorithms with stronger complexity bounds.

## 2 Sorting with Dirty Comparisons

Given a dirty predictor $\widehat{<}$, our goal is to sort $A$ with the least possible number of clean comparisons. Note that, if $\widehat{<}$ is accurate, $A$ can be sorted using only $O(n)$ clean comparisons and $O(n \log n)$ dirty comparisons. This could be achieved, for example, by performing Merge Sort with dirty comparisons and then validating the result through clean comparisons between adjacent elements. This observation motivates us to consider that not all $O(n^2)$ dirty comparisons are necessary, and we should devise an algorithm minimizing the number of both clean and dirty comparisons.

We propose a randomized algorithm that sorts $A$ with expected $O(\sum \log(2 + \eta_i))$ clean comparisons, expected $O(n \log n)$ dirty comparisons, and expected $O(n \log n)$ running time. The key idea consists of three parts: 1) Sequentially insert each element of $A$ into a binary search tree, following random order; 2) Guide each insertion primarily with dirty comparisons, while verifying the correctness of it using a minimal number of clean comparisons. 3) Correct the mistake induced by the dirty insertion, ensuring that the clean comparisons needed for correction is $O(\log \eta_i)$ in expectation.

We describe the algorithm in Section 2.1 and prove its performance guarantees in Section 2.2. In Appendix B we discuss extensions of the dirty comparison setting.

### 2.1 Algorithm

We now describe the sorting algorithm with dirty comparisons in detail (Algorithm 1). We initialize $B$ as an empty binary search tree (BST). For any vertex $v$, we denote by $\text{left}(v)$ and $\text{right}(v)$ its left and right children, and by $\text{root}(B)$ the root of $B$. Slightly abusing notation, we write $v$ also for the item stored at vertex $v$. If any of these vertices is missing, the respective variable has value NIL.

---

**Algorithm 1:** Sorting with dirty and clean comparisons

**Input:** $A = \langle a_1, \ldots, a_n \rangle$, dirty comparator $\widehat{<}$, clean comparator $<$
1   $B \leftarrow$ empty binary tree
2   **for** $i \in [n]$ in uniformly random order **do**
3      $(L_1, C_1, R_1) \leftarrow (-\infty, \text{root}(B), \infty)$
4      $t \leftarrow 1$
5      **while** $C_t \neq$ NIL **do**                         ▷ Dirty search
6          **if** $a_i \widehat{<} C_t$ **then** $(L_{t+1}, C_{t+1}, R_{t+1}) \leftarrow (L_t, \text{left}(C_t), C_t)$
7          **else** $(L_{t+1}, C_{t+1}, R_{t+1}) \leftarrow (C_t, \text{right}(C_t), R_t)$
8          $t \leftarrow t + 1$
9      $C \leftarrow C_{t^*}$, where $t^* \leq t$ is maximal s.t. $L_{t^*} < a_i < R_{t^*}$     ▷ Verification
10     **while** $C \neq$ NIL **do**                         ▷ Clean search
11          **if** $a_i < C$ **then** $C \leftarrow \text{left}(C)$
12          **else** $C \leftarrow \text{right}(C)$
13     Insert $a_i$ at $C$
14 **return** inorder traversal of $B$

---

Within each iteration of the for-loop starting in line 2, we select one item $a_i$ of $A$ uniformly randomly from the items that have not been processed yet. Then, we insert this item $a_i$ into $B$, while maintaining the invariant that $B$ remains a BST with respect to clean comparisons $<$.

Inserting item $a_i$ into $B$ requires three phases, as illustrated in Figure 2.1. The first phase involves performing a search for the insertion position using dirty comparisons $\widehat{<}$, keeping track of the search path. Here, we denote by $C_t$ the $t$th vertex on this path, and by $L_t$ and $R_t$ the lower and upper

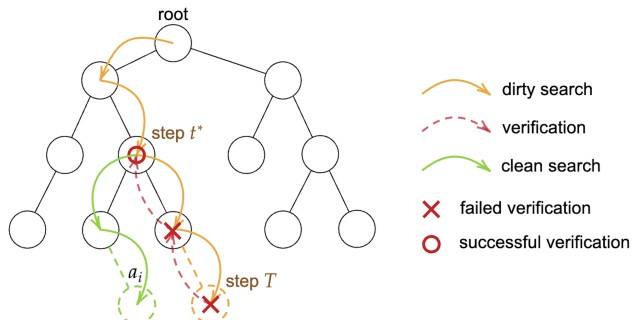

Figure 2.1: The insertion process in dirty comparison sorting.

bounds on items that can be inserted in the subtree rooted at $C_t$ without violating the BST property with respect to $<$. Correctness of the choice of $L_t$ and $R_t$ follows from the fact that $B$ was a BST with respect to $<$ before the current insertion. This dirty procedure stops when the search path reaches a NIL-leaf, regarded as the predicted position for $a_i$'s insertion. However, since we used dirty comparisons to trace the path, $a_i$ might violate one of the boundary conditions $L_t < a_i$ or $a_i < R_t$ at some recursion step $t$. We call a recursion step $t$ *valid* for $a_i$ if $L_t < a_i < R_t$. Then, we enter the verification phase in line 9. We traverse the dirty search path *in reverse order* to locate the last valid step $t^*$. A naive method to do this (which is sufficient for our asymptotic guarantees) is to repeatedly decrease $t$ by 1 until $t$ is valid; we discuss an alternate, more efficient method in Remark A.6, which yields a better constant factor.

The final phase involves performing a clean search starting from $C_{t^*}$ to determine the correct insertion position for $a_i$. After inserting $a_i$ into that position, $B$ remains a BST with respect to $<$. Once all items of $A$ are inserted into $B$, we can obtain the sorted order through the inorder traversal of $B$.

## 2.2 Complexity Analysis

The idea is to show that the dirty search path has expected depth $O(\log n)$, whereas the verification path and the clean search path only have expected depth $O(\log \eta_i)$. Therefore, the algorithm only needs $O(\log n)$ dirty and $O(\log \eta_i)$ clean comparisons for inserting each $a_i$; Theorem 1.1 is established consequently. The full proof is given in Appendix A.

## 3 Sorting with Positional Predictions

In this section, we propose two algorithms that are capable of leveraging positional predictions. The first one has complexity bounds in terms of the displacement error measure, and the second one has complexity bounds in terms of $\eta^l$ and $\eta^r$, the one-sided error measures. Each algorithm is effective in some tasks, where the given prediction is accurate with respect to its corresponding error measure.

### 3.1 Displacement Sort

We present a sorting algorithm with positional prediction, whose comparison and time complexity rely solely on the displacement error of the predictor. The algorithm is adapted from Local Insertion Sort (Mannila, 1985), an adaptive sorting algorithm proven to be optimal for various measures of presortedness.

We use a classic data structure called finger tree (Guibas et al., 1977), which is a balanced binary search tree (BBST) equipped with a pointer ("finger") pointing towards the last inserted vertex. When a new value $v$ is to be inserted, rather than searching for insertion position from the root, the insertion position is found by moving the finger from the last inserted vertex $u$ to the suitable new position. By the balance property of BBST, the insertion can be performed in $O(\log d(u, v))$ amortized time, where $d(u, v)$ is the number of vertices in the tree whose value lies in the closed interval from $u$ to $v$.

Algorithm 2 details the proposed method. We first bucket sort (in time $O(n)$) the items in $A$ based on their predicted positions, such that we may assume for all $i < j$ that $\hat{p}(i) \le \hat{p}(j)$. Following the

rearranged order, items in $A$ are sequentially inserted into an initially empty finger tree $T$. After all insertions, we obtain the exactly sorted array by an inorder traversal of $T$.

---

**Algorithm 2:** Sorting with complexity on $\eta_i^\Delta$

---

**Input:** $A = \langle a_1, \ldots, a_n \rangle$, prediction $\hat{p}$
1 BucketSort$(A, \hat{p})$;                    ▷ Bucket Sort $A$ according to $\hat{p}$, so that $\hat{p}(1) \leq \hat{p}(2) \leq \cdots \leq \hat{p}(n)$
2 $T \leftarrow$ an empty one-finger tree;
3 **for** $i = 1, \ldots, n$ **do**
4     Insert $a_i$ into $T$;
5 **return** nodes in $T$ in sorted order (via inorder traversal);

---

The full proof of run time and comparison complexity of Algorithm 2 is in Appendix C.

### 3.2 Double-Hoover Sort

Now we turn our focus to settings where one-sided errors are small. We first describe a simple algorithm with comparison complexity as a function of *either $\eta^l$ or $\eta^r$*. Following this, we introduce our algorithm Double-Hoover Sort, which has comparison complexity as claimed in Theorem 1.4. Both algorithms begin by bucket sorting $A$ with respect to the positional prediction in $O(n)$ time. Thus, we may subsequently assume that $A$ is rearranged such that $\forall i < j, \hat{p}(i) \leq \hat{p}(j)$.

**A First Approach.**    A left-sided sorting complexity of $O(\sum_{i=1}^n \log(2 + \eta_i^l))$ can be easily achieved using the standard technique of learning-augmented binary search, as described in (Lykouris and Vassilvitskii, 2021, Mitzenmacher and Vassilvitskii, 2022). Specifically, a sorted array $L$ is maintained, and $a_1, \ldots, a_n$ are sequentially inserted into $L$. During each insertion, we perform a learning-augmented binary search starting from the rightmost position of $L$, taking $O(\log(\eta_i^l + 2))$ comparisons to find the correct insertion position. In total, $O(\sum_i \log(\eta_i^l + 2))$ comparisons are taken among all insertions. By replacing the array $L$ with an appropriate data structure (e.g., a BBST with a finger that always returns to the rightmost element), one can achieve the same bound also for time complexity. A reversed "right-sided" version of this algorithm achieves complexity $O(\sum_i \log(\eta_i^r + 2))$. By simultaneously running the left-sided and right-sided algorithms, one can achieve the complexity bound of $O\left(\min\left\{\sum_{i=1}^n \log\left(2 + \eta_i^l\right), \sum_{i=1}^n \log(2 + \eta_i^r)\right\}\right)$. However, moving the $\min$ operator inside the summation requires more elaborate approach.

**Double-Hoover Sort.**    The basic idea is that, to utilize a similar insertion scheme to that employed in the one-sided algorithm, we maintain two sorted structures $L$ and $R$ at the same time, and insert each item into one of them, depending on which operation is faster, hereby achieving a complexity bound of $O(\log(\min(\eta_i^l, \eta_i^r) + 2))$. Then, a final sorted list is attained by merging $L$ and $R$ in linear time. However, a significant issue yet to be addressed is how to decide the insertion order of different items.

Consider two items $a_u$ and $a_v$ with $u < v$ (so $\hat{p}(u) \leq \hat{p}(v)$ by Bucket Sort). In our algorithm, if both are to be inserted into $L$, it is crucial that $a_u$ is inserted prior to $a_v$. Otherwise, the insertion complexity of $a_u$ could exceed the bound of $\log(\eta_u^l + 2)$. Conversely, if both $a_u$ and $a_v$ are to be inserted into $R$, then $a_v$ should be inserted prior to $a_u$. Since we cannot predict whether an item will be inserted into $L$ or $R$, formulating an appropriate insertion order that respects the constraints on both sides is impossible.

We tackle this issue of insertion order with a *strength-based, $\log n$-rounds insertion scheme*. Intuitively, we think of $L$ and $R$ as two "hoovers", with their "suction power" increasing simultaneously over time. Each item (as "dust") is extracted from the array and inserted into one hoover once the suction power reaches the required strength: a hoover with suction power $\delta$ is able to absorb items that can be inserted into it with $O(\log \delta)$ comparisons. Details are illustrated in Algorithm 3.

The sorted structures $L$ and $R$ can be implemented by arrays, though alternative data structures could provide better time complexity.

We conduct insertions in $\lceil \log n \rceil$ rounds, setting $\delta$ to be $1, 2, 4, \ldots, 2^{\lceil \log n \rceil}$. In each round, we iterate over $a_1, \ldots, a_n$ to decide if they should be inserted to $L$ in the current round. Then, we iterate over $a_n, \ldots, a_1$ reversely, to decide if they should be inserted to $R$ in the current round. Note that if an

---
**Algorithm 3:** Double-Hoover Sort
---
**Input:** $A = \langle a_1, \ldots, a_n \rangle$, prediction $\hat{p}$
1  BucketSort$(A, \hat{p})$;            ▷ Bucket Sort $A$ according to $\hat{p}$, so that $\hat{p}(1) \leq \hat{p}(2) \leq \cdots \leq \hat{p}(n)$
2  $L, R \leftarrow \langle \rangle$;
3  **for** $\delta = 2^0, 2^1, \ldots, 2^{\lceil \log n \rceil}$ **do**
4      **for** $i = 1, \ldots, n$ **if** $a_i$ *has not been inserted* **do**
5          $L^{<i} \leftarrow \{a_j \in L : j < i\}$;
6          $l_\delta^i \leftarrow$ **if** $|L^{<i}| < \delta$ **then** $-\infty$ **else** $\delta$th largest item in $L^{<i}$ ;
7          **if** $a_i > l_\delta^i$ **then**
8              Insert $a_i$ into $L$ by binary search, starting on interval $\{x \in L : l_\delta^i \leq x \leq l^i\}$,
9              where $l^i := \min_{\delta' < \delta} l_{\delta'}^i$
10     **for** $i = n, \ldots, 1$ **if** $a_i$ *has not been inserted* **do**
11         $R^{>i} \leftarrow \{a_j \in R : j > i\}$;
12         $r_\delta^i \leftarrow$ **if** $|R^{>i}| < \delta$ **then** $\infty$ **else** $\delta$th smallest item in $R^{>i}$ ;
13         **if** $a_i < r_\delta^i$ **then**
14             Insert $a_i$ into $R$ by binary search, starting on interval $\{x \in R : r^i \leq x \leq r_\delta^i\}$,
15             where $r^i := \max_{\delta' < \delta} r_{\delta'}^i$

16 **return** merge(L, R);

---

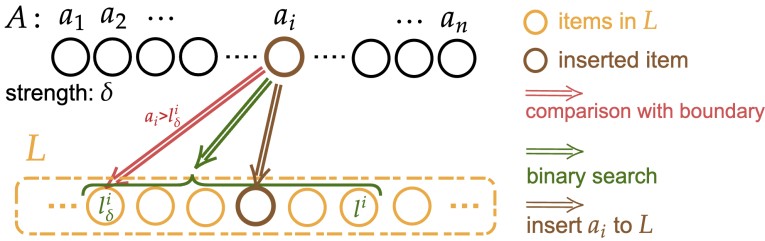

Figure 3.1: An example of the insertion process in the Double-Hoover sort.

item is inserted into either $L$ or $R$, it is omitted in later rounds. The insertion process of an item in one round is depicted in Figure 3.1.

To decide whether $a_i$ should be inserted into $L$ with strength $\delta$, let $l_\delta^i$ denote the $\delta$th largest item in $L$ with index smaller than $i$, representing the "boundary value" in this round. If there are less than $\delta$ eligible items in $L$, set $l_\delta^i$ to be $-\infty$. Let $l^i$ represent $\min_{\delta' < \delta} l_{\delta'}^i$, the minimum boundary value in previous rounds. If $l_\delta^i$ is smaller than $a_i$, we employ binary search to insert $a_i$ into $L$, starting with the interval $\{x \in L : l_\delta^i \leq x \leq l^i\}$. Conversely, to decide whether $a_i$ should be inserted into $R$, we adopt a symmetrical approach as depicted in lines 11 to 15 of Algorithm 3.

After all insertion rounds, we merge $L$ and $R$ in linear time to obtain the sorted result.

**Correctness.**    The correctness of the Double-Hoover Sort arises from the invariance that both $L$ and $R$ remain sorted after all insertions. It is sufficient to show that the initial insertion intervals at line 8 and line 14 always cover the value of $a_i$. At line 8, $l_\delta^i < a_i$ holds trivially by conditioning. Since $a_i$ is not inserted into $L$ in any previous round, $a_i < l_{\delta'}^i$ for all $\delta' < \delta$. Since the minimum operation preserves inequality, $a_i < l^i$ also holds. A similar argument can be made for the interval at line 14.

To discern the comparison complexity of the proposed algorithm, we propose the following lemma.

**Lemma 3.1.** *Upon the insertion of an item $a_i$ into $L$, all items in $L \setminus L^{<i}$ are larger than $l_i$. Similarly, when an item $a_i$ is inserted into $R$, all items in $R \setminus R^{>i}$ are smaller than $r_i$. As a result, the initial interval at line 8 and line 14 are subsets of $L^{<i}$ and $R^{<i}$, and hence have sizes no larger than $\delta$.*

**Theorem 3.2.** *For each item $a_i$, its insertion process takes $O(\log(\min\{\eta_i^l, \eta_i^r\} + 2))$ comparisons.*

The full proof of Lemma 3.1 and Theorem 3.2 are given in Appendix D.

Then, Theorem 1.4 can be obtained from Theorem 3.2, by summing up the number of comparisons in the insertion process of each $a_i$.

# 4   Experiments

In this section, we conduct experiments both on synthetic data, crafted to simulate predictions in real-world settings, and also on real-world data of countries' population ranking. The source code used for experiments is available at `https://github.com/xingjian-bai/learning-augmented-sorting`.

We assess the performance of our proposed sorting algorithms against five well-established baselines. Quick Sort and Merge Sort are classic sorting algorithms with $O(n \log n)$ complexity; Tim Sort (Peters, 2002) is a popular hybrid sorting algorithm designed to perform efficiently on real-world datasets and widely adopted in standard libraries. Further, we choose two adaptive sorting algorithms, Odd-Even Straight Merge Sort (Estivill-Castro and Wood, 1992) and Cook-Kim division (Cook and Kim, 1980), which are proven to be optimal with respect to several measures of disorderness. They serve as adaptive variants of Merge Sort and Quick Sort. To apply adaptive sorting algorithms in positional prediction settings, we first execute bucket sort on the items by their predicted ranking, breaking ties arbitrarily. This "sorted-by-prediction" array is then inputted into the baselines. In Appendix F.2, we also show results of an experimental comparison against Insertion Sort.

**Positional Predictions.**   First, we elaborate on our synthetic data generation process. In many sorting tasks, items belong to different "grades", which represent a coarse version of the ranking. For example, students are classified into grades A, B, C, and D based on their exam scores; with their grades in hand, we want to find out their accurate ranking. We denote this scenario as the *class setting*. Specifically, we divide an array of $n$ items into $c$ classes, sampling the thresholds $t_0 = 0 \le t_1 < t_2 < \ldots, t_c = n$ uniformly at random. Then, for items $a_i$ with $t_{k-1} < i \le t_k$, we say that they belong to the $k$th class, and their predicted position is uniformly generated from $(t_{k-1}, t_k]$.

To model the tasks where we have an "outdated" ranking, we design the *decay setting*. The accurate ranking is obtained as the prediction at time $0$. Then, during each time step, one item is randomly selected to be perturbed: its predicted position is shifted by $1$, towards either left or right, with uniform probability. We then ask the sorting algorithms to retrieve the original ranking of items based on the prediction at each time step.

We also utilize data from a real-world setting. We draw the annual population ranking of countries and smaller regions from 1960 to 2010 from World Bank (2023). Then, we feed in the ranking in year $x = 1960, \ldots, 2010$ respectively as the prediction, and ask the sorting algorithm to predict the ranking in year 2010.

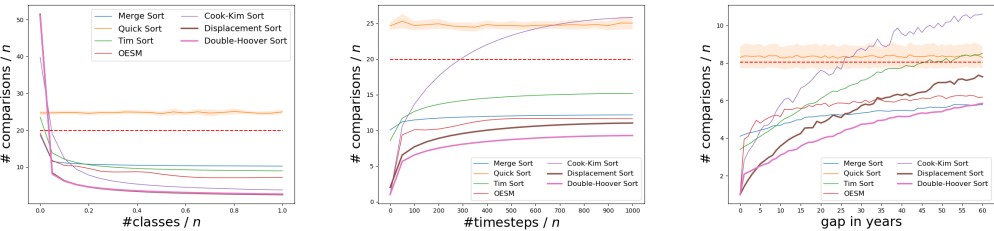

Figure 4.1: Sorting with positional predictions (left: class setting; middle: decay setting; right: country population ranking). $n = 1,000,000$ for class and decay settings, $n = 263$ for country population ranking.

In all the plots, the X-axis indicates the quality of predictions, and the Y-axis indicates the number of comparisons used. The red dotted line is $n \log_2 n$. The bold curves represent the proposed algorithms. All experiments are repeated 30 times, with the standard deviation indicated by shade. A scapegoat tree implementation of Double-Hoover Sort is used for synthetic settings, while an array implementation is used for population ranking given the small sample size.

As depicted in Figure 4.1, our algorithms consistently outperform the baselines in all settings with various task sizes. Specifically, in the class setting with $n = 1,000,000$, Displacement Sort and Double-Hoover Sort outperform all baselines when the number of classes is larger than $0.05n$. In the decay setting, both our algorithms perform better than the others as time progresses. In the real-world dataset, country population ranking, Displacement Sort needs the fewest comparisons when the given prediction is within 5 years, and Double-Hoover Sort dominates the rest when the prediction is obtained 6 to 60 years ago. These experiments illustrate that the proposed algorithms can leverage positional predictions more effectively than traditional adaptive and non-adaptive sorting algorithms in a variety of settings.

**Dirty Comparisons.**   In some sorting scenarios, some "indicating factors" can be used to cheaply compare two items. For instance, in biology, we can compare the binding affinities of two molecules for a specific target protein and provide information about their potential efficacy as drugs. However, comparisons based on indicating factors may have error induced by element-wise noise. Hence, we consider a two dirty-comparison settings in which a ratio $r$ of items is damaged. We say a dirty comparison is *perturbed* if its outcome is uniformly random. In the *Good-Dominating setting*, a dirty comparison between two items is perturbed if both are damaged; in the *Bad-Dominating setting*, a dirty comparison between two items is perturbed if either item is damaged.

In dirty comparisons settings, we use the 3-approximation feedback arc set algorithm proposed by Ailon et al. (2008) to preprocess the dirty comparisons. This algorithm uses $O(n \log n)$ dirty comparisons, the same order of magnitude as our Dirty-Clean Sort, to construct a positional prediction that roughly aligns with the given dirty comparisons. Then, we feed in the induced positional prediction to the baselines.

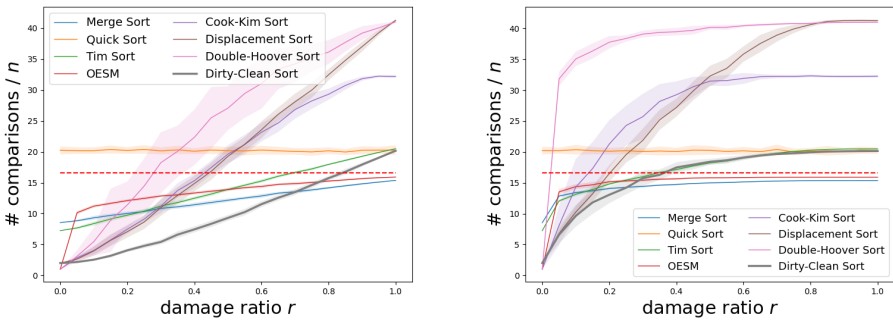

Figure 4.2: Sorting with dirty comparisons, good- and bad-dominating settings.

As showcased in Figure 4.2, when $n = 100,000$, in the Good-Dominanting setting, our proposed Dirty-Clean Sort outperforms baselines when $r \leq 0.7$; in the Bad-Dominating setting, it outperforms other algorithms when $r \leq 0.25$. If the damage ratio is large, the prediction becomes chaotic, but still performs essentially no worse than Quick Sort for reasons discussed in Remark A.6 in the Appendix.

## 5   Limitations and Future Work

In the dirty-clean setting, our algorithm still requires a time complexity of $O(n \log n)$ due to the processing of $O(n \log n)$ dirty comparisons. Consequently, the algorithm is more appropriate for situations where exact comparisons are expensive than for those where comparisons are fast. In the positional prediction setting, Displacement Sort achieves a bound of $O(\sum_i \log(\eta_i^\Delta + 2))$ for both comparison *and* time complexity, whereas the Double-Hoover sort achieves its guarantee $O(\sum_i \log(\min\{\eta_i^l, \eta_i^r\} + 2))$ only for comparison complexity. An intriguing question is whether the latter bound can be achieved for time complexity as well. Another potential limitation is that predictions might not be learnable in some sorting settings; future work could focus on exploring the conditions under which predictions are learnable.

**Broader Impact.**   Our paper proposes efficient sorting algorithms when a predictor is available. We do not expect any negative societal impact.

**Acknowledgments.** We thank the anonymous reviewers at NeurIPS and Luke Melas-Kyriazi for their valuable comments.

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

# A Proof of Comparison Complexity of Dirty-Clean Sort

The goal of this section is to prove Theorem 1.1. We start with several lemmas on the expected behavior of dirty search and clean search, focusing on a single iteration of the for-loop corresponding to item $a_i$.

For an iteration of the dirty and clean search while-loops, respectively, we call the vertex stored as $C_t$ resp. $C$ at the start of the iteration the *pivot*; the subtree rooted at the pivot is referred to as the *active subtree*. The *size* of a subtree is the number of non-NIL vertices it contains. For the dirty search, let $s_t$ denote the size of the active subtree at iteration $t$, and $T$ denote the number of recursion steps needed. Denote by $m_s^{\text{dirty}}$ and $m_s^{\text{clean}}$ the number of iterations of the dirty and clean search where the size of the active subtree lies in $(\frac{s}{2}, s]$. In particular, $m_s^{\text{dirty}} = |\{t : s_t \in (\frac{s}{2}, s]\}|$.

**Lemma A.1.** $\mathbb{E}[m_s^{\text{dirty}}] = O(1)$ *and* $\mathbb{E}[m_s^{\text{clean}}] = O(1)$ *for all $s$. Moreover,* $\mathbb{E}[m_s^{\text{dirty}} \mid s_{t^*} = s'] = O(1)$ *and* $\mathbb{E}[m_s^{\text{clean}} \mid s_{t^*} = s'] = O(1)$ *for all $s < s'$ with $\mathbb{P}(s_{t^*} = s') > 0$.*

*Proof.* We employ a percentile argument. Consider the first step of either while-loop where the active subtree has at most $s$ vertices, and let $V$ be the set of these vertices. Note that the pivot in this step is the first element in $V$ inserted into the tree. Conditioned on the vertices of the active subtree being $V$, elements of $V$ are equally likely to be the pivot, since their insertion order is uniformly random. This is true even when conditioned on $s_{t^*} = s'$, since reordering the elements within the set $V$ does not change the value of $s_{t^*}$, which is determined at higher vertices of the tree. Thus, we have at least a 50% chance that the pivot lies between the 25th percentile and the 75th percentile, in which case both children subtrees contain at most $\frac{3}{4}|V|$ vertices each. Hence, the size of active subtree shrinks by a factor of $\frac{3}{4}$ after at most 2 steps in expectation, and it shrinks to size smaller than $\frac{s}{2}$ after at most $O(1)$ steps in expectation. $\qquad\square$

Based on this lemma, we are able to characterize the expected length of dirty search, clean search, and dirty search after time $t^*$.

**Lemma A.2.** *A dirty search takes $O(\log n)$ steps in expectation, i.e. $\mathbb{E}[T] = O(\log n)$.*

*Proof.* In each dirty search, the initial largest subtree has size at most $n$. We can apply Lemma A.1 repeated on $s = n, \lfloor\frac{n}{2}\rfloor, \ldots, 1$. By linearity of expectation, we derive that a dirty search takes $\lceil \log n \rceil \cdot O(1) = O(\log n)$ steps in expectation. $\qquad\square$

**Lemma A.3.** *A clean search takes $O(\mathbb{E}[\log(s_{t^*} + 1)])$ steps in expectation; in dirty search after reaching $t^*$, there are $O(\mathbb{E}[\log(s_{t^*} + 1)])$ steps in expectation, i.e. $\mathbb{E}[T - t^*] = O(\mathbb{E}[\log(s_{t^*} + 1)])$.*

*Proof.* Assume $s_{t^*} = s'$. Then, in dirty search after $t^*$, the initial largest subtree has size $s'$, and the rest of the active subtrees all have size at most $s' - 1$. By applying the second part of Lemma A.1 on $s = s' - 1, \lfloor\frac{s'-1}{2}\rfloor, \ldots, 1$ and summing, we obtain $\mathbb{E}[T - t^*|s_{t^*} = s'] = O(\log(s' + 1))$. Thus,

$$\mathbb{E}[T - t^*] = \sum_{s'} \mathbb{E}[T - t^*|s_{t^*} = s'] \cdot \mathbb{P}[s_{t^*} = s'] = O(\mathbb{E}[\log(s_{t^*} + 1)])$$

The bound on clean search follows in the same way. $\qquad\square$

In order to relate $\mathbb{E}[\log(s_{t^*} + 1)]$ to the prediction error $\eta_i$, the next lemma first characterizes the probability of a given time step $t$ being $t^*$ as a function of the subtree size $s_t$.

**Lemma A.4.** *For any $t$ and $k$, $\mathbb{P}[t = t^* \mid s_t \in (2^{k-1}, 2^k]] \leq \eta_i/2^{k-1}$.*

*Proof.* Recall that $t^*$ is the last valid time step. A shift from a valid time step to an invalid time step occurs only if the dirty comparison between the pivot and $a_i$ is wrong. Among all $s_t$ potential pivots, at most $\eta_i$ can have mistaken dirty comparisons with $a_i$, and they are equally likely to be the pivot (depending on which of them was inserted first). Hence, given $s_t \in (2^{k-1}, 2^k]$, the probability of a pivot with mistaken comparison is at most $\eta_i/s_t \leq \eta_i/2^{k-1}$. $\qquad\square$

Based on Lemma A.4, we present the central claim bridging prediction error with comparison complexity.

**Lemma A.5.** $\mathbb{E}\left[\log\left(s_{t^*}\right)\right] = O\left(\log\left(\eta_i + 1\right)\right)$.

*Proof.* We have

$$
\begin{aligned}
\mathbb{P}\left[s_{t^*} \in (2^{k-1}, 2^k]\right] &= \sum_t \mathbb{P}\left[t = t^* \text{ and } s_t \in (2^{k-1}, 2^k]\right] \\
&= \sum_t \mathbb{P}\left[s_t \in (2^{k-1}, 2^k]\right] \cdot \mathbb{P}\left[t = t^* \mid s_t \in (2^{k-1}, 2^k]\right] \\
&\leq \mathbb{E}\left[m_{2^k}^{\text{dirty}}\right] \cdot \eta_i / 2^{k-1} \\
&\leq \eta_i \cdot O(2^{-k}),
\end{aligned}
$$

where the first inequality uses Lemma A.4 and the second is due to Lemma A.1. Thus,

$$
\begin{aligned}
\mathbb{E}[\log(s_{t^*})] &\leq \log(\eta_i + 1) \; + \; \sum_{k=\lceil \log(\eta_i+1)\rceil}^{\infty} \mathbb{P}\left[s_{t^*} \in (2^{k-1}, 2^k]\right] \cdot k \\
&\leq \log(\eta_i + 1) \; + \; \eta_i \sum_{k=\lceil \log(\eta_i+1)\rceil}^{\infty} O\left(k 2^{-k}\right) \\
&= O\left(\log\left(\eta_i + 1\right)\right). \qquad \square
\end{aligned}
$$

Theorem 1.1 is subsequently deduced.

*Proof of Theorem 1.1.* Dirty comparisons are only conducted during dirty search, when the recursion step is incremented by one. As per Lemma A.2, the total number of dirty comparisons is bounded by the sum of steps across all insertions, which is $n \cdot O(\log n)$.

In each verification phase, as we traverse the dirty search in reverse order to locate $t^*$, at most $T - t^* + 2$ clean comparisons suffice. In each clean search phase, the expected number of clean comparisons is the expected number of steps in clean search. Therefore, based on Lemma A.3 and Lemma A.5, the total number of clean comparisons performed in both phases is $O\left(\sum \log\left(\eta_i + 2\right)\right)$.

Additionally, the running time is dominated by the number of dirty and clean comparisons. $\qquad \square$

**Remark A.6.** The algorithm can be implemented such that the number of clean comparisons is at most that of quicksort plus $O(n \log \log n)$, regardless of prediction error. Thus, even with terrible predictions our algorithm matches the performance of quicksort up to a factor that tends to 1 as $n \to \infty$.

To achieve this, we can implement the verification step by decreasing $t$ in geometrically increasing step sizes until a valid $t$ has been found, and then perform a binary search for $t^*$ between the last two attempted values of $t$. This reduces the number of clean comparisons in line 9 from $O(T - t^*)$ to $O(\log(T - t^*))$, which is at most $O(\log \log n)$ in expectation by Lemma A.2, and at most $O(n \log \log n)$ for all verification steps together. The remaining clean comparisons are performed during the clean searches. In the worst case (when all dirty comparisons are incorrect) all clean searches start from the root, and together they perform exactly the same set of comparisons as quicksort (by a coupling argument between the random choices of the two algorithms: E.g., the root of the search tree corresponds to the initial uniformly random pivot of quicksort).

# B   Extentions of Dirty Comparison Setting

We consider two extensions of the dirty comparison setting. The first one assumes that dirty comparisons are probabilistic; the second one discusses the setting where multiple predictors are available. We briefly discuss the extension of our algorithms and results in these new settings.

## B.1 Probabilistic Dirty Comparisons

Our algorithm extends to the case where dirty comparisons are probabilistic. Assume that for each pair of objects $i, j$, the dirty comparison between them yields an incorrect result with probability $\eta_{ij}$. Algorithm 1 can be directly applied in this setting, achieving the same guarantees by defining $\eta_i = \sum_j \eta_{ij}$. The proof remains unchanged.

If repeatedly querying the same dirty comparison multiple times yields independent results, the number of clean comparisons can be further reduced: Let $\epsilon_{ij} := \min\{\eta_{ij}, 1/2\}$. When querying a dirty comparison $2k$ times, the probability that the correct answer fails to secure a majority vote is at most $(4\epsilon_{ij}(1 - \epsilon_{ij}))^k$: For $\eta_{ij} \geq 0.5$, this bound is trivial. Otherwise, there are $2^{2k}$ strings of length $2k$ over the alphabet {correct, incorrect}, and each string that is at least half incorrect has probability at most $(\epsilon_{ij}(1 - \epsilon_{ij}))^k$.

So by repeating each dirty comparison query $2k$ times, we obtain an algorithm that performs $O(kn \log n)$ dirty comparisons and $O\left(\sum_i \log\left(\sum_j (4\epsilon_{ij}(1 - \epsilon_{ij}))^k\right)\right)$ clean comparisons.

## B.2 Multiple Predictors

We now discuss the setting where multiple predictors are available and prove Theorem 1.2. Suppose we have $k$ different dirty comparison predictors. Let $\eta_i^p$ denote the number of incorrect comparisons by predictor $p$ for item $i$.

We prove Theorem 1.2 by reduction to the problem of "prediction with expert advice": In this problem, there are $k$ experts, and each incurs a loss in the range $[0, 1]$ per time step. An algorithm must select an expert in each round before the losses are revealed and then incurs the loss of the chosen expert. According to (Freund and Schapire, 1997, Equation(9)), their algorithm HEDGE has an expected loss of $O(L + \log(k))$, where $L$ is the total loss of the best expert in hindsight.

In our case, the experts correspond to the predictors, and time steps correspond to the $n$ iterations of the for-loop of Algorithm 1 where an item is inserted into the BST. We define the loss of expert $p$ in the time step where $a_i$ ought to be inserted by $\ell_i^p = \log(1 + \tilde{\eta}_i^p)/\log(n)$, where $\tilde{\eta}_i^p \leq \eta_i^p$ is the number of incorrect comparisons of predictor $p$ between $a_i$ and the items already in the BST at this time. Note that the value of $\tilde{\eta}_i^p$ can be determined at the end of the time step (once $a_i$ is correctly placed in the BST) without any additional clean comparisons; the division by $\log n$ ensures that $\ell_i^p \in [0, 1]$ as required.

The algorithm for multiple predictors proceeds as Algorithm 1, querying for dirty comparisons at a given time step the predictor $p$ corresponding to the expert chosen by HEDGE at that time step. Recall from the analysis in Section 2.2 that the expected number of clean comparisons in this time step is then $O(\log(1 + \tilde{\eta}_i^p))$, which is an $O(\log n)$ factor larger than the loss suffered by HEDGE. Consequently, by the $O(L + \log k)$ bound on the cost of HEDGE, the total expected number of clean comparisons is $O(\min_p \sum_i \log(1 + \eta_i^p) + \log(k) \log(n))$. Since $k \leq 2^{O(n/\log n)}$, the term $\log(k) \log(n) = O(n)$ is negligible, and Theorem 1.2 follows.

## C  Proof of Comparison Complexity of Displacement Sort

*Proof of Theorem 1.3.* We focus on the insertion process of each item $a_i$ in Algorithm 2. Let $d_i$ denote the number of nodes between $a_i$ and $a_{i-1}$ in an inorder traversal of the tree after inserting $a_i$, including themselves. These nodes must have their correct ranking in the final sorted list between $p(i - 1)$ and $p(i)$; hence

$$d_i \leq |p(i) - p(i - 1)| + 1, \text{ for all } i = 2, \ldots, n.$$

Therefore, the running time and number of comparisons among all insertions are bounded by

$$\sum_{i=2}^{n} O(\log(d_i)) \leq \sum_{i=2}^{n} O(\log(|p(i) - p(i-1)| + 1))$$

$$\leq \sum_{i=2}^{n} O(\log(|p(i) - \hat{p}(i)| + |p(i-1) - \hat{p}(i-1)| + \hat{p}(i) - \hat{p}(i-1) + 1))$$

$$\leq \sum_{i=2}^{n} O(\log(3 \cdot \max\{\eta_i^\Delta + 1, \eta_{i-1}^\Delta + 1, \hat{p}(i) - \hat{p}(i-1) + 1\}))$$

$$\leq O(n) + \sum_{i=1}^{n} O(\log(\eta_i^\Delta + 1))$$

$$\leq O\left(\sum_{i=1}^{n} \log(\eta_i^\Delta + 2)\right).$$

The second inequality is by $\hat{p}(i-1) \leq \hat{p}(i)$ and the triangle inequality; the third inequality is by monotonicity of logarithm. The penultimate inequality is justified by $\log(\max\{x, y, z\}) \leq \log x + \log y + \log z$ for all $x, y, z \geq 1$ and

$$\sum_{i=2}^{n} \log(\hat{p}(i) - \hat{p}(i-1) + 1) \leq \sum_{i=2}^{n} (\hat{p}(i) - \hat{p}(i-1)) \leq n.$$

This concludes the proof of Theorem 1.3. $\qquad \square$

## D  Proof of Comparison Complexity of Double-Hoover Sort

**Lemma D.1.** *Upon the insertion of an item $a_i$ into $L$, all items in $L \setminus L^{<i}$ are larger than $l_i$. Similarly, when an item $a_i$ is inserted into $R$, all items in $R \setminus R^{>i}$ are smaller than $r_i$. As a result, the initial interval at line 8 and line 14 are subsets of $L^{<i}$ and $R^{<i}$, and hence have sizes no larger than $\delta$.*

*Proof.* Assume $a_i$ is inserted into $L$ in round $\delta$. Consider any $a_j \in L \setminus L^{<i}$ at the time of $a_i$ insertion. Then, $a_j$ was inserted into $L$ previously with a smaller insertion strength $\delta' < \delta$. By the insert condition, $a_j > l_{\delta'}^j$.

Since $L^{<i}$ is a subset of $L^{<j}$, the $\delta'$th largest item of the latter set must exist and be no smaller than the $\delta'$th largest items of the former set. Then, we obtain

$$a_j > l_{\delta'}^j \geq l_{\delta'}^i \geq l^i.$$

Hence, the interval $\{x \in L | l_\delta^i \leq x \leq l^i\}$ only contains items in $L^{<i}$. Since there are $\delta$ items in $L^{<i}$ that are no smaller than $l_\delta^i$, the interval has at most $\delta$ items. An analogous proof shows the symmetric property in $R$. $\qquad \square$

**Theorem D.2.** *For each item $a_i$, its insertion process takes $O(\log(\min\{\eta_i^l, \eta_i^r\} + 2))$ comparisons.*

*Proof.* Each item $a_i$ goes through some rejected insertions in earlier rounds, and then gets inserted into $L$ or $R$ in a certain round. We refer to them as the *exploration phase* and *insertion phase*, respectively, and prove that the number of comparisons needed in both phases is bounded by $O(\log(\min\{\eta_i^l, \eta_i^r\} + 2))$.

First, we claim that each $a_i$ is inserted into either $L$ or $R$ prior to or during the round with insertion strength $\delta_i^l := 2^{\lceil \log(\eta_i^l + 2) \rceil}$. If $a_i$ is inserted before round $\delta_i^l$, the claim trivially holds. The claim also holds if at round $\delta_i^l$, $L^{<i}$ contains fewer than $\delta_i^l$ items. In the absence of these conditions,

$$\left|\{a_j \in L^{<i} : a_j > a_i\}\right| = |\{a_j \in L : j < i \wedge a_j > a_i\}|$$

$$\leq |\{j \in [n] : \hat{p}(j) \leq \hat{p}(i) \wedge p(j) > p(i)\}|$$

$$= \eta_i^l < \delta_i^l.$$

Hence, in round $\delta_i^l$, $a_i$ must be larger than the boundary value, which is the $\delta_i^l$th largest item in $L^{<i}$. Consequently, it will be inserted into $L$ in round $\delta_i^l$.

A similar argument shows that $a_i$ will be inserted prior to or during round $2^{\lceil \log(\eta_i^r + 2)\rceil}$. Combining these two bounds, we find that $a_i$ must be inserted prior to or during round $\delta_i := 2^{\lceil \log(\min\{\eta_i^l, \eta_i^r\} + 2)\rceil}$. Hence, the exploration phase only needs $O(\log(\min\{\eta_i^l, \eta_i^r\} + 2))$ comparisons.

Next, we continue to examine the number of comparisons needed in the insertion phase. Suppose $a_i$ is inserted into $L$ in some round $\delta \le \delta_i^l$. Then, the binary search starts with an interval of size

$$|\{x \in L: l_\delta^i \le x \le l^i\}| \le \delta \le \delta_i^l = O(\min\{\eta_i^l, \eta_i^r\} + 2).$$

The first inequality is due to Lemma 3.1. Hence, the insertion phase of $a_i$ by binary search needs $O(\log(\min\{\eta_i^l, \eta_i^r\} + 2))$ comparisons.

$\square$

# E    Lower Bounds on Comparison Complexity

We prove the lower bounds stated in Theorem 1.5.

## E.1    Optimality of Displacement Sort

In this section, we show that an exact sorting algorithm, augmented with a positional prediction, cannot sort the array with $o(\sum_{i \in [n]} \log(\eta_i^\Delta))$ comparisons.

**Definition E.1.** Given $n$, a positional prediction $\hat{p}$, and a real number $U$, define the size of the $U$-candidate set as

$$\mathrm{cand}(\hat{p}, U) := \left| \left\{ A \in S_n : \sum_{i=1}^{n} \log(\eta_i^\Delta + 2) \le U \right\} \right|,$$

where $S_n$ is the set of permutations of $[n]$ (viewed as an array), $\eta^\Delta$ is calculated accordingly for each $A$ against $\hat{p}$.

**Theorem E.2.** *Given any function $f(U) = o(\max_{\hat{p} \in S_n} \log(\mathrm{cand}(\hat{p}, U)))$, there does not exist any positional augmented sorting algorithm with comparison complexity $O(f(U))$ for instances with $\sum_{i=1}^{n} \log(\eta_i^\Delta + 2) \le U$.*

*Proof.* Given the predictor $\hat{p}$ and an upper bound $U$ on the error, a sorting algorithm needs to determine the correct permutation $A$ from a candidate set of size $\mathrm{cand}(\hat{p}, U)$. If the algorithm only uses $x$ comparisons, then it can only distinguish $2^x$ different outcomes; if $2^x$ is smaller than the number of candidates, it cannot determine the correct answer in every situation, since the information given by comparisons is not sufficient to distinguish all candidates. Hence, at least $\lceil \log \mathrm{cand}(\hat{p}, U) \rceil$ comparisons are needed. $\square$

**Theorem E.3.** *For all $n \le U \le O(n \log n)$, we have $\max_{\hat{p} \in S_n} \log \mathrm{cand}(\hat{p}, U)) = \Omega(U)$.*

*Proof.* Proof by construction. Assume without loss of generality that $U$ is a multiple of $n$ and let $U' = U/n$. Take $\hat{p} = \langle 1, 2, \dots, n \rangle$, the identity prediction. We aim to construct sufficient number of candidate permutations $A \in S_n$, which all fall in the $U$-candidate set.

Consider every permutation $A \in S_n$ constructed in the follow way: Initially, set $A = \langle 1, 2, \dots, n \rangle$. Then, divide $A$ into $\frac{n}{2^{U'}}$ adjacent subarrays, each containing $2^{U'}$ items (the last subarray is potentially smaller if there are not enough remaining elements). Finally, we permute the items in each subarray arbitrarily, and retrieve $A$ after the permutation.

In each possible outcome $A$, $\log \eta_i^\Delta \le U'$ for all $i$ since each item is only permuted locally. Hence, the error of $\hat{p}$ w.r.t. $A$ is no larger than $n \cdot U' = U$. Each $A$ falls inside the $U$-candidate set. Counting the number of possible different outcomes in our construction, we obtain

$$\mathrm{cand}(\hat{p}, U) \ge \left[ \left( 2^{U'} \right)! \right]^{\frac{n}{2^{U'}}},$$

where the right-hand-side represents the possible ways to permute $2^{U'}$ items in each subarray.

By Stirling's formula,

$$\log \text{cand}(\hat{p}, U) \geq \Omega \left( \frac{n}{2^{U'}} \cdot (2^{U'} \cdot U') \right) = \Omega(U).$$

$\square$

Directly combining Theorem E.2 and Theorem E.3, and noting that at least $\Omega(n)$ comparisons are always needed to verify correctness of a sorted list, we obtain the following:

**Corollary E.4.** *For sorting with positional predictions, there exists no algorithm with comparison complexity $o(\sum_{i=1}^{n} \log(\eta_i^\Delta + 2))$.*

### E.2 Optimality of Dirty Comparisons Sort and Double-Hoover Sort

We can use the same definition of $\text{cand}(\hat{p}, U)$ and construction as above. Every positional prediction can be view as a total linear relation on array $A$, therefore induces a unique dirty-comparison predictor. Since each element is locally perturbed, we can prove that for each permutation $A$ obtained from the construction, $\sum_{i=1}^{n} \log(\eta_i + 2)$ and $\sum_{i=1}^{n} \log(\min \{\eta_i^l, \eta_i^r\} + 2)$ are also upper bounded by $U$. Hence, the optimality of Algorithm 2 and 3 can be proven in the exact same way, and Theorem 1.5 follows.

## F Extra Experimental Results

### F.1 Synthetic Datasets with Varying Sizes

This section presents additional experimental results to provide a more comprehensive evaluation of our proposed algorithms. As scalability is a crucial aspect, we conducted further experiments on synthetic datasets with varying sizes from n = 1,000 to n = 1,000,000. Results are shown in Figure F.1, F.2, and F.3.

In these figures, the shaded area denotes the standard deviation of each run among 30 repetitions. As the results show, our proposed algorithms perform consistently with relatively small deviations.

### F.2 Comparisons against Insertion Sort

The performance of Insertion Sort against our proposed algorithms is shown in Figure F.4. As the plots show, insertion sort runs fast for small errors in the decay and country settings, though the performance declines steeply when predictions deteriorate (i.e., smoothness is worse, and robustness is non-existent). The experimental results suggest that in practical settings, it is a good idea to incorporate insertion sort as a fall-back option in case predictions are extremely accurate.

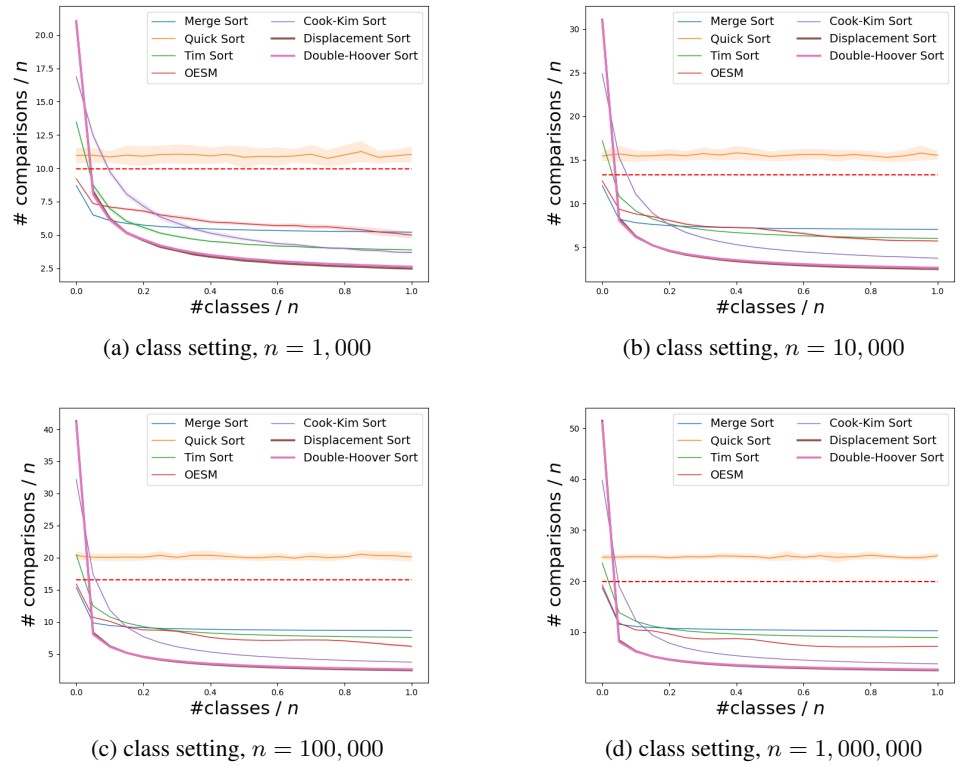

(a) class setting, $n = 1,000$

(b) class setting, $n = 10,000$

(c) class setting, $n = 100,000$

(d) class setting, $n = 1,000,000$

Figure F.1: Class Settings.

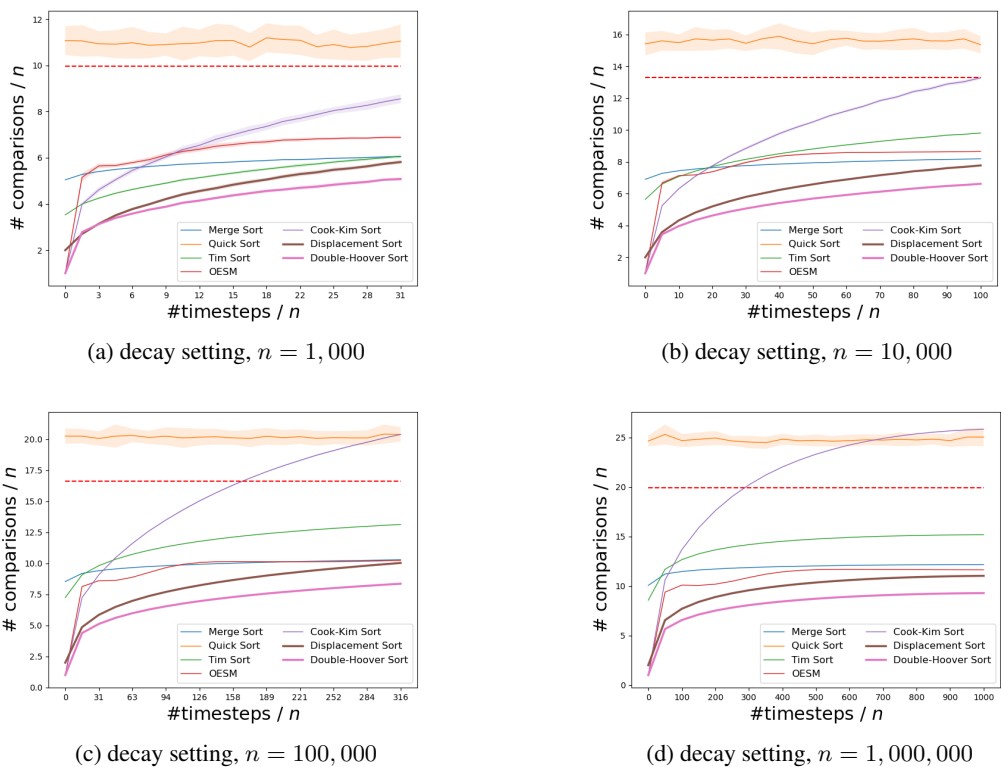

(a) decay setting, $n = 1,000$

(b) decay setting, $n = 10,000$

(c) decay setting, $n = 100,000$

(d) decay setting, $n = 1,000,000$

Figure F.2: Decay Settings.

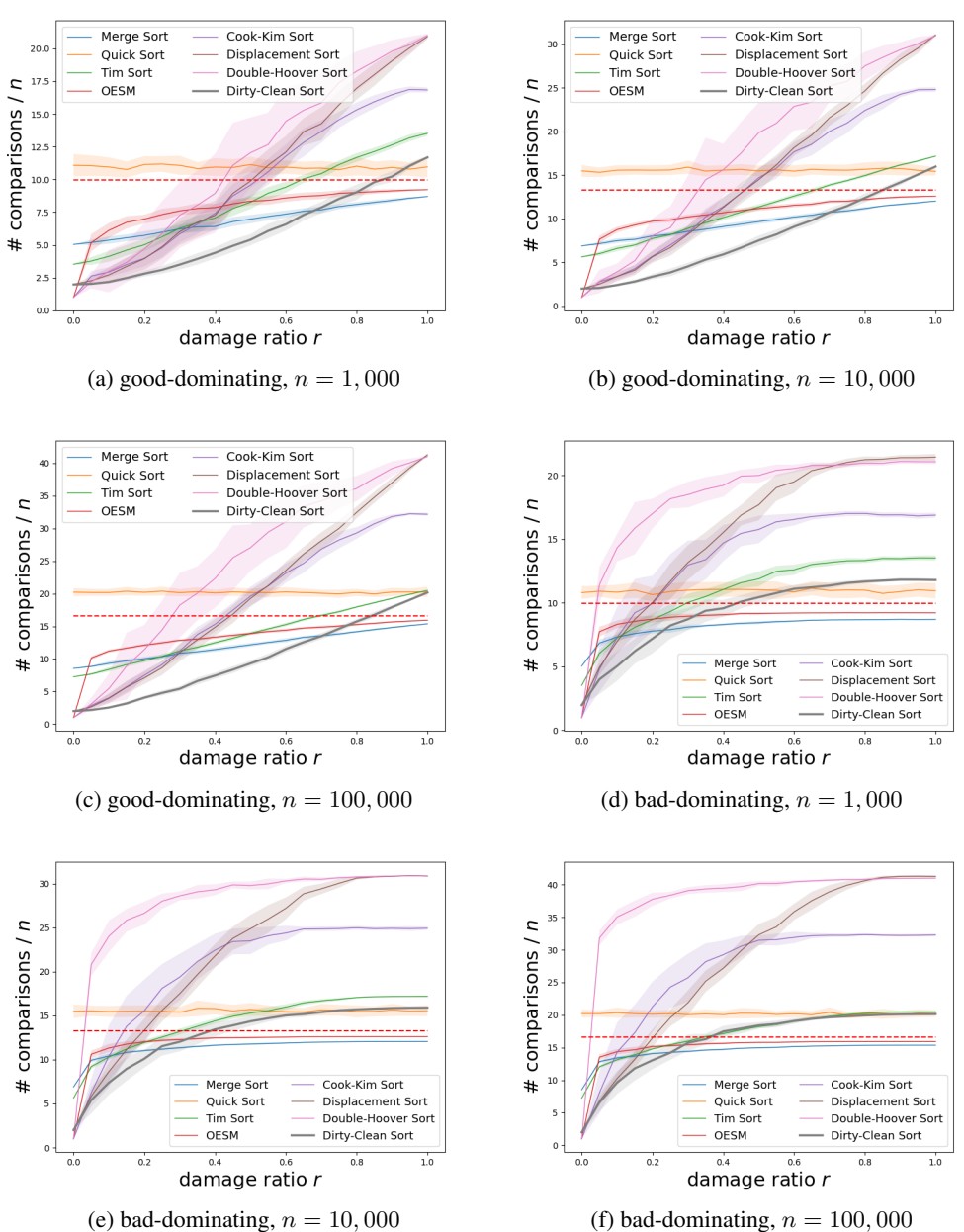

(a) good-dominating, $n = 1,000$

(b) good-dominating, $n = 10,000$

(c) good-dominating, $n = 100,000$

(d) bad-dominating, $n = 1,000$

(e) bad-dominating, $n = 10,000$

(f) bad-dominating, $n = 100,000$

Figure F.3: Sorting with dirty comparisons, good- and bad-dominating settings

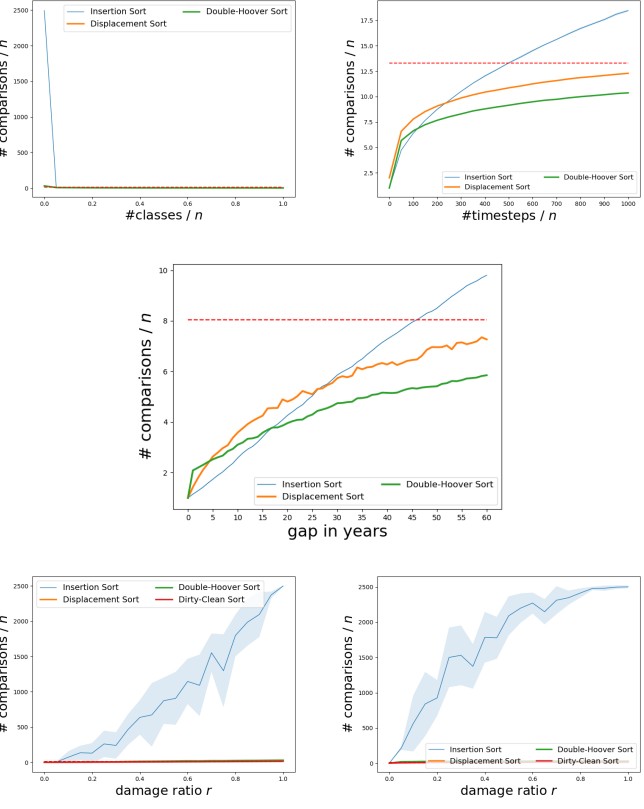

(a) Results of insertion sort in positional and dirty-comparison settings. Due to the $O(n^2)$ complexity of comparisons, $n = 10,000$ (rather than $1,000,000$ as in the main text) is used in the class, good-dominating, and bad-dominating settings.

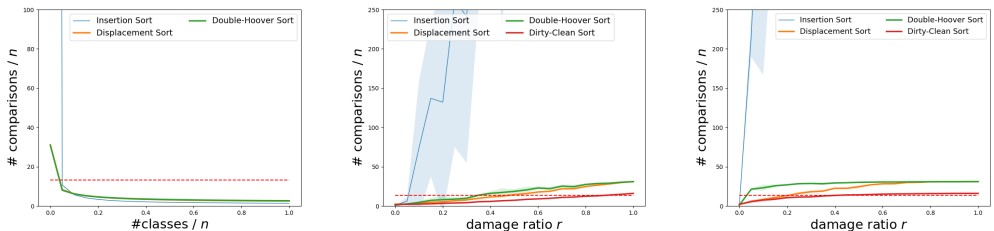

(b) Zoomed-in plots for class setting, good- and bad-dominating settings.

Figure F.4: Comparison with Insertion Sort

