# OpenReview forum: "Sorting with Predictions"
_NeurIPS.cc/2023/Conference — NeurIPS 2023 poster_

### Official Review · Reviewer_gjeT · 2023-06-23

**Soundness:** 4 excellent
**Presentation:** 4 excellent
**Contribution:** 3 good
**Rating:** 7
**Confidence:** 3

**Summary:**

This paper presents sorting algorithms augmented by prediction under two settings: (i) sorting with positional predictions and (ii) sorting with dirty and clean comparisons. In (i), the algorithm can use a predicted position $\hat{p}(i)$ of each element $i$, and in (ii), the algorithm is provided a faster but possibly inaccurate comparison oracle besides a slow but accurate comparison. In both settings, the authors present provably faster algorithms making use of the positional predictions/dirty comparisons. They also show that the complexities of the presented algorithms are asymptotically optimal. The efficacy of the algorithms is validated through numerical experiments on synthetic and real-world datasets.

**Strengths:**

The two settings introduced in this paper are well-grounded in real-world scenarios, demonstrating their relevance and applicability. The algorithms proposed are elegant and straightforward, contributing to their practical utility. I also find the lower-bound results commendable. The paper is well-composed, particularly the introduction, which provides an accessible entry point for readers unfamiliar with algorithm-with-predictions research. Therefore, my overall assessment leans towards a positive evaluation of this paper.

**Weaknesses:**

- The sorting algorithm with dirty comparisons is randomized and takes $O(n \log n)$ comparisons, as explained in Section 5. The contribution of this paper will be stronger if the algorithm is derandomized and/or the worst-case time-complexity bound is improved to $O(\sum_i \log (\eta_i^\triangle + 2))$.
- The algorithms cannot be implemented in the "in-place manner" because they use extra binary trees.

**Questions:**

- The height of a binary search tree can be deterministically maintained within a logarithmic order of #nodes by using the "rotation" operation. Can the presented algorithm with dirty comparisons also be derandomized by rotating the tree?
- What happens if there are ties in the correct order of elements? Are your sorting algorithms work? If so, are they stable? Can you extend your result to the setting where the dirty comparison for two elements reports that "they are in the same order"?
- The insertion sort runs fast if the array is almost sorted. Thus, in the positional prediction setting, the bucket sort (by the prediction) followed by the insertion sort seems to run fast. I'm interested in an experimental comparison.

**Limitations:**

Yes

---

> ### Author Rebuttal · Authors · 2023-08-09
>
> We thank the reviewer for their positive feedback and interesting comments. We address specific comments below.
>
> - Derandomizing the dirty comparison algorithm is an interesting question, and we do not know how to do this. Indeed the height of the tree can be made O(log n) deterministically via rotations. But the main purpose of randomization is to ensure that the root of each subtree is a uniformly random element from that subtree, so that the probability of an incorrect dirty comparison is at most $\eta_i$ divided by the number of elements in that subtree. For example, if there is just a single bad element that has incorrect dirty comparisons with all other elements, while other dirty comparisons are correct, then one would have to prevent an adversary from tricking a deterministic algorithm into placing that bad element at the root. We don’t know how to do this, but it is a nice question for future research.
> - Improving the running time of the dirty comparison algorithm to match the number of clean comparisons isn’t possible: Even if all dirty comparisons are correct, the total number of (dirty or clean) comparisons needs to be at least $n\log n$ due to the lower bounds on classical sorting.
> - All our algorithms work also if there are elements of equal value (essentially, this only helps the algorithms to run faster; we disregarded this case in the write-up in the interest of simplicity of presentation). If stability is desired, this could be achieved by adding a tie-breaker that compares, whenever a comparison turns out equal, the positions of the two items in the input array (this may seem like a hack, but it does not affect the asymptotic guarantees). It is also possible to allow dirty comparisons to output “equal”, and count this as an error of 1/2 if they are non-equal. Also here, the same guarantees can be achieved (idea: any equal dirty comparison can be interpreted as random).
> - The question regarding comparison with Insertion sort is addressed in our general response to all reviewers.

---

### Official Review · Reviewer_HoTq · 2023-06-28

**Soundness:** 3 good
**Presentation:** 3 good
**Contribution:** 3 good
**Rating:** 6
**Confidence:** 3

**Summary:**

This paper studies learning-augmented sorting, where an algorithm obtains access to a potentially erroneous prediction on the sorting of an array. The paper considers two prediction models. In the positional predictor, every item has a prediction for its place in the sorted array, and the error of item i is the number of indices its predicted location is from its true location in the sorted array. The also consider a one sided error model for that prediction. In sorting with dirty comparisons, every pair of items has a < or > relation between them. Note that the dirty comparisons do not have to satisfy transitivity, thus potentially giving rise to cycles. A query between two elements is called a clean comparison.

The authors design sorting algorithms that use these predictions, and the performance of the algorithms depends on the error in the prediction. The performance is measured by the number of comparisons. Two of the main results (given with matching lower bounds) are:
- Given access to dirty comparisons, there is a randomized alg sorting in O(n log n) runtime, O(n log n) dirty comparisons, and O(sum_i log(eta_i + 2)) clean comparisons in expectation.
- Given a positional predictor, there is an alg sorting in O(sum_i log(|p(i)-\hat{p}_i|+ 2)) runtime and comparisons.

**Strengths:**

- The paper is well written, and I believe the theoretical results to be correct.
- The experimental results sufficiently back the theory (for my taste).
- Overall, I like the paper well enough. However, I’m concerned about how much new insight into sorting with predictions this is contributing, given that similar problems have been studied and the authors do not thoroughly compare to these prior works. Can the authors comment more on this?


**Weaknesses:**

- While the paper gives a nice background on learning-augmented algorithms, I think several key works are missing that need to be compared against. For example, I need a thorough comparison against The Case for a Learned Sorting Algorithm by Kristo et al. Also, the papers The Case for Learned Index Structures by Kraska et al and SageDB: a Learned Database System by Kraska et al should be at least mentioned. I would also like a few more sentences explaining why the Lu et al and Kristo et al algorithms are not amenable to this setting/ what the notable differences are.
- From a cursory glance, I believe the paper from Kristo et al has analysis that depends on the total error of the indices, whereas the paper here considers a more fine-grained analysis by depending on each indices error. I don’t know how/ if the algorithms are different. Can the authors please comment more on this?

**Questions:**

(Please see questions in Weaknesses and Strengths)

Many grammatical typos throughout, I highlights a few below but I recommend a full read through before submitting a CR to Neurips or submitting to another venue. Here are some things I noticed below:
- (lines 15-17) The first sentence in the intro is oddly phrased, consider rewording.
- (lines 24-38) Add citations for the examples of how learning-augmented sorting could be useful.
- (line 41) In *the* dirty comparison setting…
- (line 42) comparison *bounds* on the error…
- (line 126-127) While *the* recurrent noisy…
- (Lines 176-179) I think slightly more detail here could be added. The point is that with the dirty comparisons you end up without a total ordering, so the point is the go to the closest subtree that contains something breaking the total ordering and then fix that element’s position in this subtree? So this clean search is what will remove the cycles.
- (Line 293) We design *the* decay setting.

---

> ### Author Rebuttal · Authors · 2023-08-09
>
> We thank the reviewer for their feedback and in particular pointing out additional related works. We inspected these works more closely, and while they may seem similar at first glance, they actually consider different problems, and the techniques from these papers cannot be used to conclude any of our results, as discussed below. Nonetheless, we shall indeed expand the discussion of related works in our paper to better contrast it from prior work and avoid misunderstandings. We hope this will address the reviewer’s concerns.
>
> First, we wonder whether there may have been some misunderstandings about our paper, since the reviewer’s summary mentions that our algorithm for positional predictions makes dirty comparisons, although these are two completely separate settings (when given positional predictions, we do not make any dirty comparisons). There also appears to be some mix-up regarding running times in the review’s summary of results.
>
> We are confused by the reviewer’s statement that “the problem has already been studied”. We are not aware of any works that study the classical sorting problem in a setting where the algorithm’s input is augmented with predictions. Granted, the paper of Lu et al. that we cite studies *generalized* sorting with predictions, but for the non-generalized setting they do not obtain any non-trivial guarantees (details below). The positional prediction model is similar to adaptive sorting, as we mention in the paper, though the perspective via predictions motivates the more refined error measures (and this also leads to better experimental performance compared to popular adaptive sorting algorithms).
>
> **Comparison with Kristo et al.:** The paper of Kristo et al. proposes an algorithm for sorting numerical values that performs well in empirical tests. In the first step, their algorithm tries to approximate the empirical CDF of the entire input by applying ML techniques to a small subset of the input. However, although the algorithm has a learning component, it does not really fall into the area of “algorithms with predictions” since the input to the algorithm is just the list of items (no additional predictions). Without providing additional information to the algorithm’s input (via e.g. predictions), it is not possible to sort faster than $O(n \log n)$ on average due to the entropic lower bound (unless one makes other assumptions, e.g. numeric inputs drawn from an efficiently approximable CDF, or input values belonging to a bounded range of integers). The reviewer states “From a cursory glance, I believe the paper from Kristo et al has an analysis that depends on the total error of the indices”. We do not find any such analysis in the paper of Kristo et al. As far as we see, the paper never defines an error measure – please let us know in case we missed something. The paper is mostly empirical, whereas ours is mostly theoretical. That said, Kristo et al. does contain a brief theoretical discussion, but it appears the theoretical complexity of their algorithm is $O(n^2)$ due to an insertion sort step, though they argue that empirically it runs much faster. Indeed, this makes sense for numerical inputs drawn from a sufficiently nice distribution, since then one can extrapolate from a small part of the input to the rest. If inputs are non-numeric (and no monotonous mapping to numbers is known), then one has to rely on a comparison function, and the approach of Kristo et al. would not be well-defined. In contrast, our algorithms can sort arbitrary types of items since we only assume access to a comparison function and predictions (the latter being necessary to beat $O(n \log n)$). We do not see any significant similarities between our algorithms and theirs, and already the settings are entirely different. That said, we acknowledge that we should have discussed their work due to a similar high-level motivation.
>
> **More details on Lu et al.:** As mentioned in our paper, this work studies the so-called “generalized sorting” problem with predictions. In generalized sorting, some comparisons are forbidden and others are allowed, and the goal is to sort the array by making only allowed comparisons (assuming this is possible). The paper considers the setting where for each allowed comparison, a prediction on the outcome is available (which is similar to our dirty comparisons, so we shall use the terminology of dirty and clean comparisons hereafter). They provide two algorithms whose numbers of clean comparisons are $O(n \log n + w)$ and $O(nw)$ respectively, where $w$ is the total number of incorrect dirty comparisons. The running time in both cases is polynomial (the exact polynomial is not stated). These bounds are interesting when a significant portion of comparisons are forbidden, since then it is not known how to sort faster than $O(n^{1.5})$ without predictions. But note that in the non-generalized setting where *all* comparisons are allowed, the first bound is never better than what can already be achieved without predictions. The second bound could be better than $O(n \log n)$ if predictions are very good, but already when each item is involved in only a single incorrect and otherwise correct dirty comparison, the $O(nw)$ bound becomes $O(n^2)$, whereas ours is $O(n)$. Adapting these algorithms to our dirty comparison setting would yield an exponentially worse error dependence compared to our algorithms, and the ideas in this paper would not be suitable to obtain any of our results.
>
> We will expand the discussion on related work in the final version of our paper, including also the other (mostly empirical) works by Kraska et al. The provided references along with the other feedback will help us improve our paper. However, we do not see a concern that our work might be close to others’ validated.

---

> > ### Comment · Reviewer_HoTq · 2023-08-15
> >
> > Thanks to the authors for these detailed overviews of these priors works.
> >
> > First off, I understand the two settings of the of the positional predictors vs the dirty comparisons.  I think there was just a typo on my part when I was summarizing the theorem statements (my apologies!).
> >
> > Secondly, I believe your discussion on prior work has assuaged my concerns on the novelty of this work. I had been aware that there was work on sorting (while not falling into the algorithm with predictions framework perfectly) that seemed similar from a high-level, and I was just confused why it wasn't discussed further. Indeed, the authors from the Kristo et al paper call their algorithm a "ML-enhanced sorting algorithm", and it's from at least some of the same authors that wrote the very influential paper in the algorithms with predictions space The Case for Learned Index Structures. However, “the problem has already been studied” was too flippant of a statement from myself and I will retract that in an edit of my review. All I was looking for was a more complete understanding of how your work compares to those works. I now understand how those works compare to yours. In particular, this part of your response on the work by Kristo et al. helped me understand why their ideas could not be used to get your results: "Indeed, this makes sense for numerical inputs drawn from a sufficiently nice distribution, since then one can extrapolate from a small part of the input to the rest. If inputs are non-numeric (and no monotonous mapping to numbers is known), then one has to rely on a comparison function, and the approach of Kristo et al. would not be well-defined. In contrast, our algorithms can sort arbitrary types of items since we only assume access to a comparison function and predictions (the latter being necessary to beat )". I additionally appreciated the discussion on the Lu et al. paper, as again I now understand why their techniques could not extend to your setting!
> >
> > I highly recommend adding some of this discussion as space allows. I will be updating my review to reflect a more positive view on this paper's acceptance.

---

> > > ### Author Response · Authors · 2023-08-20
> > >
> > > Thank you, we much appreciate the response and the support for our paper.

---

### Official Review · Reviewer_S6Cj · 2023-07-03

**Soundness:** 4 excellent
**Presentation:** 4 excellent
**Contribution:** 3 good
**Rating:** 8
**Confidence:** 5

**Summary:**

The paper studies the oracle-augmented sorting problem, in which, in addition to the array, an oracle is provided to return ‘reference answers’ for the sorting of the array. The oracle is presumably efficient – queries are assumed to take $O(1)$ time – but might be noisy, i.e., at times return incorrect answers. The key question is to design algorithms that 1). When the oracle is great, the number of comparisons is significantly smaller than the baseline algorithms; and 2). When the oracle is very bad, the number of comparisons is still comparable to the baseline algorithms.

To this end, the paper considered oracles under two settings: a). An oracle that gives ‘dirty comparisons’ – a list of ‘all-pair’ comparisons that may include some errors and b). An oracle that gives a positional list – a list of predicted positions for each element in the array. The paper then designs algorithms under both models. In particular, for the dirty comparison model, the proposed algorithm achieves $O(\sum_{i=1}^{n} \log \eta_{i})$ number of true comparisons and $O(n \log n)$ dirty comparisons, where $\eta_i$ is the number of ‘wrong comparisons’ based on element $a_i$. This demonstrates a smooth tradeoff between the number of true comparisons needed and the prediction error. The paper also obtained similar results for positional list predictions under different notions of error.

I read the dirty-clean comparison algorithm and proofs to some details, and they look correct as far as I can tell. I skimmed the claims for the positional list algorithms, but did not verify the proofs (these algorithms seem to have built on more recent ideas). For the dirty-clean comparison, I think a key observation here is that the binary comparison trees are fairly ‘robust’ against perturbations in the sense that 1). The error is consistent – if there is $R<L$ on level $t$, then the error also happens for all $t’>t$ and 2). Thanks to randomization, the probability of the mess-up of the upper and lower bounds becomes exponentially small when the level moves up. As such, the binary search tree should be able to travel sufficiently deep in the tree based only on the dirty comparisons unless a very large chunk of the comparisons is corrupted. These observations, although not technically novel, do seem to be neat and cute for this problem.

Overall I think the paper makes a significant contribution to a fundamental problem, and the setting is interesting and well-motivated. The paper is also well-written and easy to follow; the experiments also demonstrate competitive performance for the new algorithms. As such, I have a positive evaluation of the paper.


**Strengths:**

As I mentioned in the general evaluation, I think the paper is in a nice shape both in terms of the contributions and the presentations. To elaborate a bit more:
- The paper follows a recent line of work on ‘learning-augmented algorithm’/’algorithm with advice’, and it tackles a very important and fundamental problem. It is surprising that previous work has not considered the problem.
- Although the techniques are not complicated, the algorithms are based on several neat observations. The upper bounds are also tight by matching the lower bounds (unfortunately deferred to the appendix).
- In addition to the theoretical results, the experimental performance is also quite competitive – one interesting observation is that the dirty-clean sort remains very competitive to baseline even when the corruption rate approaches 1. Furthermore, the experiments are run on large scales, which verifies the efficiency also from the practical time side (not necessarily captured by the theoretical time complexity).

**Weaknesses:**

- One potential criticism of the work is that the techniques are not novel and perhaps easy to come up with. This is true for the dirty-clean sorting algorithm; and although I *personally* do *not* think this is an issue, this perspective is frequently flagged for theoretical work. The positional list with left-right errors seems to require more involved techniques, but I do not really like that definition of error.
- It seems the two settings of the oracles in this paper are very natural. However, there could still be other settings that your model does not capture. For instance, one can define a dirty comparison oracle with amortized error only (across ${n \choose 2}$ comparisons), and what will be the optimal bound therein? That being said, I don’t think it’s very fair to ask a paper to address all possible models, so I’m not letting this question affect my evaluation.
- Some (minor) presentation issues: In the binary tree search subroutines of algorithm 1, the use of tuples makes the text slightly confusing. I struggled for a minute to parse the value assignment of tuples. Maybe you can say we use tuples to represent (lower-bound, pivot, upper-bound) before pointing the readers to the algorithm. Also, what does it mean for ‘return inorder traversal of B’? (I know what it means, but is this the best way to say it?)


**Questions:**

- In your clean-dirty comparison model, I think you have this ‘deterministic error’ that if the comparison between a and b is wrong, you cannot get the correct answer by querying it again. This crucially separates your model from the classical noisy comparison line of work (your ‘dirty perturbation’ is not random). If what I said is correct, I would suggest adding a discussion about this.
- In your experiments, the dirty-clean sorting seems extremely robust even when the damage ratio approaches $1$. Is this an artifact of your experiment setting/your implementation, or is there a reason that the algorithm performs in such a nice way? (Note that your proof shows only *asymptotically* competitive, but it seems even the constant is very competitive when the damage ratio is high.)
(More presentation-related)
- Does the term BST on page 4 stand for ‘binary search tree’? The abbreviation was never mentioned before.
- Line 185, it’s a bit weird to talk about ‘timestep’ here – I think you mean the recursion step.


**Limitations:**

Most of the limitations are discussed in the ‘weakness’ section. From a broader societal perspective, since the work is of theoretical nature, there is no immediate negative impact, and I could not foresee any (take it with a grain of salt since I do not have experience in ethical/societal review).

---

> ### Author Rebuttal · Authors · 2023-08-09
>
> We thank the reviewer for their positive feedback and interesting questions and observations.
>
> **Simplicity:** We believe that the simplicity of our algorithms is a strength as it makes them more widely appreciable, especially given that this is the initial investigation of the subject.
>
> **Amortized error:** Assuming that “amortized error” refers to the sum or average of the $\eta_i$, then our algorithms do in fact achieve optimal bounds in terms of this measure as well. Please see our response to reviewer JwAo under the heading “Other error measures”.
>
> **Noisy sorting:** The distinction between the dirty-clean model and noisy sorting as observed by the reviewer is correct. Moreover, the noisy model does not allow *exact* sorting, which we overcome by allowing clean comparisons. We will expand our existing discussion on noisy sorting to clarify this. In our general response to all reviewers, we discuss how our algorithms can also be extended to probabilistic dirty comparisons (note some dirty comparisons might have error probability $\ge 1/2$, which noisy sorting doesn’t allow).
>
> **Good-looking constant factors in experiments:** Indeed, a slightly more careful analysis shows that even if dirty comparisons are arbitrarily bad, the expected number of clean comparisons is at most that of quicksort plus $O(n\log\log n)$ (and in particular, the leading $n\log n$ term in the bad-prediction case has the same constant factor as for quicksort). This crucially relies on what we mention in the footnote on page 4, which makes the verification step negligible (a naive implementation of the verification step would lose a factor of 2). In the worst case where all dirty comparisons are incorrect, all clean searches start from the root. But then all clean searches together perform exactly the same set of comparisons as quicksort (by a coupling argument between the random choices of the two algorithms: E.g., the root of the search tree corresponds to the initial uniformly random pivot of quicksort). This explains why even with terrible predictions, dirty-clean sorting matches the performance of quicksort up to a factor that tends to 1 for large n.
>
> We thank the reviewer for additional suggestions that we will incorporate in the final version (in particular, yes, BST = binary search tree).

---

> > ### Comment · Reviewer_S6Cj · 2023-08-13
> > **Response**
> >
> > Thanks for providing answers to my queries. I have read both the review rebuttal and the general rebuttal. I think the authors did a fairly good job of explaining additional details and results under slightly different models. I strongly encourage the authors to include them in later versions (from my side, I think it is definitely worth highlighting that the constant factor in your dirty-clean sorting is $2$ in the worst case, and the overhead can be made $n \log\log n$).
> >
> > I also skimmed my colleagues' reviews and comments. I do believe work among the line of 'algorithms with prediction' is suitable for NeurIPs. I'm also fairly familiar with the literature therein, and it appears the sorting with prediction problem is not studied before.
> >
> > I'm not *entirely* dropping the technicality point: to be clear, I love simple and cute results, but I don't think this paper contains an 'aha' type of idea. I still think the paper is very positive despite this -- I'm only using the technicality point to distinguish award-quality work vs. others.
> >
> > A final question: By reading your response on ``other error measures'',  it's unclear whether the bound is optimal from my end. If I understand it correctly, defining $D=\sum_{i} \eta_{i}$, your upper bound is $n \log(\frac{D}{n})$; in contrast, your lower bound is $\Omega(\sum_{i=1}^{n} \log \eta_{i})$. Log is a concave function so you have $n \log(\frac{D}{n}) \geq \sum_{i=1}^{n} \log \eta_{i}$, no? So it seems we can't simply the bound is tight, unless I missed something.

---

> > > ### Author Response · Authors · 2023-08-14
> > >
> > > We appreciate the additional comments. To clarify why we claim tightness of the bound with respect to $D$: The idea is that our lower bound proof bounds each $\eta_i$ by the same quantity, so Jensen’s inequality is essentially tight for the constructed family of instances. More precisely, the same lower bound proof can be adapted to the alternative error definition with only the following minor modifications: In the definition of the $U$-candidate set, replace $\sum_i\log(\eta_i)$ by $n\log(D/n)$. Then replace lines 530-531 by the observation that $\eta_i\le 2^{U’}$, hence $D\le n2^{U’}$, hence $n\log(D/n)\le nU’=U$.

---

> > > > ### Comment · Reviewer_S6Cj · 2023-08-14
> > > > **Increasing socres by 1**
> > > >
> > > > Thanks for the clarification. I do not have any further questions. Btw, I increased both my evaluation and the confidence score by 1 to reflect a more firm opinion after the rebuttal discussions.

---

> > > > > ### Author Response · Authors · 2023-08-20
> > > > >
> > > > > Thank you, we much appreciate the strong support.

---

### Official Review · Reviewer_JwAo · 2023-07-08

**Soundness:** 3 good
**Presentation:** 3 good
**Contribution:** 3 good
**Rating:** 4
**Confidence:** 4

**Summary:**

The paper proposes algorithms for sorting a set of items when unreliable extra information is available in the form of positional predictions or comparisons. The paper proposes a randomized algorithm for the former and deterministic algorithms for the latter problem.

**Strengths:**

The paper proposes interesting algorithms for sorting when additional unreliable information is available.
The algorithms are simple and intuitive and the analysis (though several ideas are standard) is effective.


**Weaknesses:**

- A potential weakness is the relevance to the learning (NeuRIPS) community. In my opinion, the paper is more suited to a theoretical computer science venue as opposed to an ML venue such as NeuRIPS. Would like to understand the author's view on this


**Questions:**

- Will it help to detect at some point in the algorithm that the dirty comparisons are going to hurt more than help i.e., if the algorithm can learn/find out that it will spend more comparisons (dirty + clean) than just the O(nlogn) clean comparisons, it might abandon the dirty comparisons and thus help in reducing the total number of comparisons. Is this possible in the suggested algorithms? Is there a learning component that is natural here?
- Standard metrics for comparing two permutations include the Kendall-Tau distance (between permutations), pairwise disagreement error (permutation vs tournament), etc. It might be useful to state the results in terms of these metrics.
- How do these algorithms change when one has a budget of clean comparisons to work with?
- An important pairwise comparison setup in the machine learning context is when the comparisons are not deterministic but probabilistic - for instance when a clean comparison is made between items i and j, the result might be based on the toss of a coin with probability P_{ij} = s_i/ (s_i + s_j) where s_k is the score of item k (The Bradley-Terry-Luce model). Here, one needs to learn the score vector to be able to rank (or sort) the items. How easy/hard is it to extend the results in this paper where one needs to be able to learn these scores from dirty or clean comparisons - if yes, this will make the work much more relevant to the ML community.


**Limitations:**

Listed above in the weakness section - the results are less suited to a ML venue in my opinion.

---

> ### Author Rebuttal · Authors · 2023-08-09
>
> We thank the reviewer for their comments and interesting suggestions. We believe we can address the perceived potential weakness of suitability for NeurIPS, and provide answers to this and other questions below.
>
> **Suitability to the NeurIPS community:** We recognize that our paper is not a classical ML paper, but the call for papers is explicitly broad in encouraging submissions from a wide range. For the area of “algorithms with predictions” (that our paper belongs to), NeurIPS and ICML have actually been the main venues: In fact, there have been more papers from this area at NeurIPS alone than at the classical theory/algorithms venues STOC, FOCS, SODA, ICALP and ESA combined. (There is a website tracking papers from the area to verify this, but we are not allowed to provide links here.) Machine Learning is often a natural way of generating predictions, and while there are some works that prove this rigorously, it requires distributional assumptions. Since powerful predictions can often be obtained even without provable guarantees (via ML or otherwise), the majority of these theoretical works focus on how to utilize predictions rather than how to generate them. Some examples just from last year’s NeurIPS alone include Bernardini et al. “A Universal Error Measure for Input Predictions Applied to Online Graph Problems”, Grigorescu et al. “Learning-Augmented Algorithms for Online Linear and Semidefinite Programming”,  and Jin and Ma “Online Bipartite Matching with Advice: Tight Robustness-Consistency Tradeoffs for the Two-Stage Model”. Given a history of many more similar works at NeurIPS in recent years, we believe that it is the ideal venue for our paper.
>
> **Suggestion of completely abandoning dirty comparisons if they are detected unhelpful:** This is indeed possible via existing learning techniques, which learn online which of two algorithms performs best. However, it would not improve the theoretical guarantees: Note that even with arbitrarily bad predictions, the total number of dirty + clean comparisons of our algorithm is never worse than $O(n \log n)$, since $\eta_i\le n$.
>
> **Other error measures:** If positional predictions are bijective (so that Kendall-Tau distance is well-defined), then Kendall-Tau is equal (up to a factor 2) to $\sum_i \eta_i$. Similarly, in the dirty-clean setting, pairwise disagreement error is equal to $\sum_i \eta_i$. Writing $D$ for either of these measures, the guarantees of our algorithms are $O(n\log(D/n))$ by Jensen’s inequality, and this is optimal as a function of $D$ (our same lower bound construction applies). Note, however, that our more refined bounds are stronger. We will expand the related discussion currently in lines 67-73 and move it to a more prominent place in the final version.
>
> **Budget of clean comparisons:** This is an interesting idea. A budget less than what our algorithm achieves wouldn’t allow to *exactly* sort the input, due to our lower bounds. However, it is conceivable that one could prove interesting results in such a setting if the goal is to approximately sort the input, or if dirty comparisons are random and it can help to query the same dirty comparison multiple times.
>
> The question about probabilistic dirty comparisons, as well as an additional result about combining multiple predictors via learning techniques, are discussed in our general response to all reviewers.

---

> > ### Comment · Reviewer_JwAo · 2023-08-20
> >
> > Thanks for the response. I acknowledge that I have read the rebuttal.

---

> > > ### Author Response · Authors · 2023-08-20
> > >
> > > We thank the reviewer for getting back to us. The response also leaves us somewhat puzzled though. We understood that the primary concern seemed to have been the relevance to the NeurIPS community, which we thought we addressed by pointing to (a small fraction of) the multitude of works of the same research area that appeared at NeurIPS in recent years. Though the reviewer's response (and maintenance of their score) makes us wonder whether they still believe that NeurIPS is an unsuitable venue, or if there is some other point that we may have missed. We note that some of the questions are answered in our general response rather than the reviewer-specific rebuttal.
> > >
> > > Granted, we of course cannot claim that our work aligns with the research interests of all parts of the community. This appears to be the case with every paper, from all areas. NeurIPS's interdisciplinary ambition led to many works on "algorithms with predictions" being accepted in recent years, including also as spotlight and oral presentations. We highlight that reviewer S6Cj's answer also addressed the raised concern, stating, “I do believe work among the line of 'algorithms with prediction' is suitable for NeurIPs.”

---

### Author Rebuttal · Authors · 2023-08-09

We thank all reviewers for their valuable feedback and many interesting questions and suggestions. We provide answers in individual responses. It appears the main concerns besides otherwise mostly positive feedback were whether our work is suitable for the NeurIPS community (reviewer JwAo) and that the relationship with prior work wasn’t sufficiently discussed (reviewer HoTq), which may have additionally led to a mistaken impression that the problem was already studied. We believe that we can address both concerns, as outlined in the individual responses. Should there be more questions though, we would be happy to provide further information.

Additionally, this general answer addresses three points that might be of interest to all reviewers:

**Probabilistic dirty comparisons:** Reviewer JwAo asked whether our algorithms extend to settings with probabilistic dirty comparisons. A first answer is that one can apply our algorithm as is, achieving the same guarantee when defining $\eta_i=\sum_j \eta_{ij}$, where $\eta_{ij}$ is the probability that dirty comparison between i and j is incorrect. The proof is completely unchanged. If repeated queries to the same dirty comparison give independent results, one can do better: Let $\epsilon_{ij} := \min(\eta_{ij}, 1/2)$. Then, querying this dirty comparison 2k times, the probability that the correct answer fails to win the majority vote is at most $(4\epsilon_{ij}(1-\epsilon_{ij}))^k$. (For $\eta_{ij} \ge 1/2$ this is trivial. Otherwise, there are $2^{2k}$ length-2k-strings over the alphabet {correct,incorrect}, and each at-least-half-incorrect string has probability at most $(\epsilon_{ij}(1-\epsilon_{ij}))^k$.) So by reduction to our first answer to this question, we obtain an algorithm with $O(k n\log n)$ dirty comparisons and $O(\sum_i \log(\sum_j (4\epsilon_{ij}(1-\epsilon_{ij}))^k))$ clean comparisons. More sophisticated algorithms are conceivable that choose different k for different pairs $(i,j)$. Also, the problem where each dirty comparison costs 1, clean comparisons cost $M>1$, and the goal is to minimize the total cost becomes interesting in the probabilistic case. We will add a discussion to our paper.

**Multiple predictors:** Reviewer JwAo’s questions about strengthening the links to Learning further motivate us to include an additional result in our paper. It is about a setting with $k$ different dirty comparison operators (aka predictors), where the best operator is unknown. Let $\eta_i^p$ be the number of wrong comparisons of predictor p for item i. We claim that as long as $k\le \exp(n / \log n)$, we can sort with at most $O(\min_p \sum_i \log(2+\eta_i^p) )$ clean comparisons, i.e., as good as if we knew in advance which predictor was best. This follows via reduction to the problem of “prediction with expert advice”: There are $k$ experts, each suffering a loss in [0,1] per time step. An algorithm has to choose an expert in each round before losses are revealed, and then suffers the loss of the chosen expert. By [Freund and Schapire. A Decision-Theoretic Generalization of On-Line Learning and an Application to Boosting. Equation (9)], there is an algorithm with expected loss $O(L+\log(k))$, where $L$ is the total loss of the best expert. In our case, the experts correspond to the predictors, and the loss of expert $p$ in round $i$ can be defined as $\log(1+\eta_i^p) / \log(n)$ so that it lies in [0,1] as required (to be more precise, since knowing $\eta_i^p$ would require many clean comparisons, this should be replaced by the according quantity restricted to the set of items that were already inserted into the BST). So loss is a $1 / \log(n)$ fraction of number of clean comparisons when using predictor $p$ to guide the insertion of item i. Then the total number of clean comparisons is $O(\min_p \sum_i \log(\eta_i^p) + \log(k)\log(n))$, where the term $\log(k)\log(n) = O(n)$ is negligible for $k\le \exp(n / \log n)$. The bound on $k$ is essentially tight since already $k=\exp(n\log n)$ would mean there could be one predictor for each of the n! possible sorting outcomes, which would render them useless.

**Experimental comparison with insertion sort:** Reviewer gjeT asked for an experimental comparison with bucket sort (according to predictions) followed by insertion sort. The reviewer’s expectation that this runs fast for small errors is correct in some settings (and indeed, in some experiments, there’s a range where it outperforms our algorithms for small error), though the performance deteriorates steeply when predictions get worse (i.e., smoothness is worse, and robustness non-existent). Plots with results are attached. This suggests that for practical settings it is a good idea to incorporate insertion sort as a fall-back option in case predictions are extremely good.

---

### Decision · Program_Chairs · 2023-09-21

**Decision:**

Accept (poster)

**Comment:**

The authors are encouraged to incorporate parts of their response in the paper. In particular, they should:
* Elaborate on the tightness claim.
* Incorporate, as extensions, the settings of probabilistic dirty comparisons, multiple predictors
* Add the experimental results with insertion sort.
* Highlight connections with/differences from the works by Kristo et al. and Lu et al.
* Implement the numerous smaller improvements (typos, presentation, etc.)